# Quantifying Cross-Attention Interaction in Transformers for Interpreting TCR-pMHC Binding

[1]**Jiarui Li,** [1]**Zixiang Yin,** [2]**Haley Smith,**
[1]**Zhengming Ding,** [2]**Samuel J. Landry,** [1*]**Ramgopal R. Mettu**
[1] Department of Computer Science, Tulane University
[2] Department of Biochemistry and Molecular Biology, Tulane University School of Medicine
`{jli78, zyin, hsmith19, zding1, landry, rmettu}@tulane.edu`
[*]Corresponding author.
**https://qcai.jiarui.li/**

## Abstract

CD8+ "killer" T cells and CD4+ "helper" T cells play a central role in the adaptive immune system by recognizing antigens presented by Major Histocompatibility Complex (pMHC) molecules via T Cell Receptors (TCRs). Modeling binding between T cells and the pMHC complex is fundamental to understanding basic mechanisms of human immune response as well as in developing therapies. While transformer-based models such as TULIP have achieved impressive performance in this domain, their black-box nature precludes interpretability and thus limits a deeper mechanistic understanding of T cell response. Most existing post-hoc explainable AI (xAI) methods are confined to encoder-only, co-attention, or model-specific architectures and cannot handle encoder-decoder transformers used in TCR-pMHC modeling. To address this gap, we propose Quantifying Cross-Attention Interaction (QCAI), a new post-hoc method designed to interpret the cross-attention mechanisms in transformer decoders. Quantitative evaluation is a challenge for XAI methods; we have compiled TCR-XAI, a benchmark consisting of 274 experimentally determined TCR-pMHC structures to serve as ground truth for binding. Using these structures we compute physical distances between relevant amino acid residues in the TCR-pMHC interaction region and evaluate how well our method and others estimate the importance of residues in this region across the dataset. We show that QCAI achieves state-of-the-art performance on both interpretability and prediction accuracy under the TCR-XAI benchmark.

## 1 Introduction

T cells play a pivotal role in the adaptive immune system by identifying and responding to antigenic proteins, both from pathogens such as viruses, bacteria and cancer cells (Joglekar & Li, 2021) as well as in the context of autoimmunity. The final and arguably most critical component of T cell response is binding between the peptide Major Histocompatibility Complex (pMHC) which contains an antigenic peptide bound to a MHC molecule and the surface receptor on T cells (TCR). The specificity of this interaction underpins T cell-mediated immunity and is an intense area of research in both the development of therapies and fundamental understanding of immune response. Understanding T cell response is the key to vaccines that confer long-lasting immunity, and can also enable effective personalized cancer therapies (Rojas et al., 2023; Poorebrahim et al., 2021).

Transformer models have recently been use to analyze T cell immunity (Hudson et al., 2023; Li et al., 2023; Karthikeyan et al., 2023; Driessen et al., 2024; Cornwall et al., 2023). Specifically, models have been developed to predict TCR-pMHC binding such as TULIP (Meynard-Piganeau et al., 2024), Cross-TCR-Interpreter (Koyama et al., 2023), TCR-BERT (Wu et al., 2024b), and

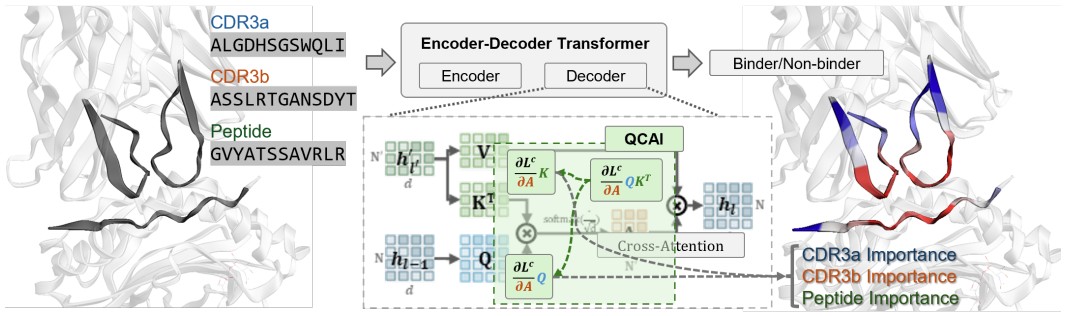

Figure 1: **Quantifying Cross-Attention Interaction (QCAI)** is a post-hoc explanation method designed for cross-attention mechanisms. In this paper, we show that QCAI enables insight into the structural basis for TCR-pMHC binding.

BERTrand (Myronov et al., 2023)[1]. However these models are black boxes and suffer from a lack of interpretability, which is critically important in elucidating the mechanisms involved in T cell response. To address this challenge, post-hoc explanation techniques (Kenny et al., 2021) have been developed to connect elements of the input and model to the outputs. However, current existing post-hoc methods (e.g., AttnLRP (Achtibat et al., 2024), and TokenTM (Wu et al., 2024a)) are designed for encoder-only transformer or convolution neural network (CNN) models, while state-of-the-art TCR-pMHC binding predictors adopt encoder-decoder architectures.

The main contribution of this paper is to fill this gap with a novel post hoc explanation method that we call **Quantifying Cross-Attention Interaction (QCAI)** that enables interpretation of any encoder-decoder transformer model while taking cross-attention into account. Our motivation is the application of TCR-pMHC modeling, but cross-attention is used extensively in vision and NLP applications as well and thus QCAI has the potential to be applied to other fields in Appendix A.12. Figure 1 shows how QCAI is used to analyze decoder blocks; cross-attention between the CDR3 and peptide sequences is captured used to generate importance scores by residue position. Another important contribution of this paper is to provide a way to quantify the performance of XAI methods for TCR-pMHC binding. Typically XAI methods are evaluated qualitatively (e.g. in image analysis), but in the context of immunology, interpretations that match intuition are challenging to justify. We introduce **TCR-XAI**, a compilation of 274 experimentally-determined X-ray crystallography structures of TCR-pMHC complexes. For each complex we determine interaction distance between the CDR3 regions and peptide. Using this benchmark, we can determine whether the importance scores over the input produced by any particular method matches the expected interaction shown in the experimental structure.

We performed extensive evaluation of QCAI against other post hoc methods and demonstrate the benefit of incorporating cross-attention. We conduct an extensive comparison with other methods over the TCR-XAI benchmark and demonstrate that QCAI achieve state-of-the-art performance. We also analyze two case studies of TCR-pMHC systems to highlight mechanisms identified by QCAI.

## 2 PRELIMINARIES

In this section we first outline some basic concepts for self-attention and cross-attention and then discuss limitations in existing post-hoc methods. Transformer-based architectures typically consist of $L$ stacked encoder layers, or a combination of encoder and decoder layers. Each layer comprises two primary components: Multi-Head Attention (MHA) and a Feed-Forward Network (FFN), each followed by layer normalization and residual connections (Vaswani et al., 2017). The distinction between encoder and decoder modules lies in their input structure and the type of attention mechanism employed.

In the encoder, each layer takes the output of the previous layer $h_{l-1} \in \mathbb{R}^{N \times d}$ and computes $h_l \in \mathbb{R}^{N \times d}$, where $N$ is the number of tokens and $d$ is the hidden dimension. In contrast, the decoder layer integrates two inputs: $h_{l-1} \in \mathbb{R}^{N \times d}$ from the previous decoder layer, and $h'_l \in \mathbb{R}^{N' \times d}$ from

---

[1]For a comprehensive discussion, please consult Section A.4 of the Appendix.

the corresponding encoder layer. The decoder output remains $h_l \in \mathbb{R}^{N \times d}$, with $N'$ denoting the number of source tokens.

These inputs are linearly projected into query $(Q_l)$, key $(K_l)$, and value $(V_l)$ matrices for the MHA computation. For encoder, it could be computed following $Q_l = W_l^Q h_{l-1}$, $K_l = W_l^K h_{l-1}$, and $V_l = W_l^V h_{l-1}$. For decoder, it could be computed following $Q_l = W_l^Q h_{l-1}$, $K_l = W_l^K h_l'$, and $V_l = W_l^V h_l'$. Where $W_l^Q, W_l^K, W_l^V \in \mathbb{R}^{d \times d}$ are trainable projection matrices. For brevity, bias terms are omitted. Considering a single attention head for simplicity, the attention matrix $A_l$ for layer $l$ is computed as:

$$A_l = \mathrm{softmax} \left( \frac{Q_l K_l^\top}{\sqrt{d}} \right).$$

The shape of $A_l$ is $\mathbb{R}^{N \times N}$ for the encoder and $\mathbb{R}^{N \times N'}$ for the decoder. The output of the attention module is computed as: $h_l = W_l^O \left( A_l V_l + h_{l-1} \right) \in \mathbb{R}^{N \times d}$, where $W_l^O \in \mathbb{R}^{d \times d}$ is a learnable output projection matrix. Outputs from multiple attention heads are concatenated before being linearly transformed.

## 2.1 LIMITATIONS OF CURRENT INTERPRETABILITY METHODS FOR TRANSFORMERS

Several post-hoc interpretability methods, such as TokenTM (Wu et al., 2024a), AttnLRP (Achtibat et al., 2024), and AttCAT (Qiang et al., 2022), have demonstrated reliable performance on encoder-only transformer models that rely on self-attention. However, these methods are not designed to extract the interaction information from cross-attention found in decoder layers. As a result, their applicability is limited in models that include decoder components, such as TULIP (Meynard-Piganeau et al., 2024) and MixTCRpred (Croce et al., 2024).

The core distinction between self-attention and cross-attention lies in the source of the key $(K)$ and value $(V)$ matrices. While self-attention derives $Q$, $K$, and $V$ from the same input, cross-attention uses separate inputs for $Q$ and $(K, V)$. Consequently, the attention matrix $A$ in cross-attention has dimensions $\mathbb{R}^{N \times N'}$ instead of $\mathbb{R}^{N \times N}$, where $N$ is the number of query tokens and $N'$ is the number of key tokens. Additionally, $A$ now represents the fused information from both modalities. This asymmetry poses a challenge for interpretability: the attention matrix no longer provides a direct measure of query token importance of one input modality, making it difficult to attribute model predictions to input query tokens.

## 3 QUANTIFYING CROSS-ATTENTION INTERACTION

In this section we present our main contribution, which is a way to handle the aforementioned asymmetry so that cross-attention can be captured. Since the attention matrix is computed as a scaled dot-product $QK^\top$, which captures the cosine similarity between query and key representations, interpreting the cross-attention mechanism can be structured into three key steps: 1. identifying which components of the attention matrix contribute most significantly to the model's prediction, 2. decomposing these importance values into contributions from the query and key inputs, respectively, and 3. aggregating the cross-attention explanation with other layers' explanation.

Inspired by GradCAM (Selvaraju et al., 2017), we propose to compute the importance of the attention matrix $A_l$ at layer $l$ using the gradient of the loss $L^c$ with respect to $A_l$ for a target class $c$, in conjunction with the attention weights themselves. Specifically, we define the importance score map as:

$$\mathbf{S}(A_l) = \mathbb{E}_H \left( \mathrm{ReLU} \left( \frac{\partial L^c}{\partial A_l} \odot A_l \right) \right) + I \in \mathbb{R}^{N \times N'},$$

where $\mathbb{E}_H(\cdot)$ denotes averaging across all attention heads, $\odot$ represents element-wise multiplication, and $I$ denotes the identity matrix for residue connection. This formulation highlights the attention entries that both have high weights and contribute significantly to the class-specific loss. The next step is to quantify this attention importance map into contributions from the query and key inputs. By analyzing the structure of the attention matrix, which serves as a soft alignment between queries and keys, we aim to attribute the importance scores back to the input tokens in both sequences.

### 3.1 Quantifying Query Importance from Cross-Attention

For the query input $Q_l$ at layer $l$, its importance scores with respect to the loss $L^c$ for class $c$ can be estimated in a GradCAM-style fashion as:

$$\mathbf{S}(Q_l) = \text{ReLU}\left(\frac{\partial L^c}{\partial Q_l} \odot Q_l\right),$$

where $\odot$ denotes element-wise multiplication. To obtain token-level importance scores from this matrix, we compute the column-wise maximum:

$$\omega_l^Q = \arg\max_i \mathbf{S}(Q_l)_{i,j} \in \mathbb{R}^N,$$

where $i$ indexes the feature dimension, $j$ indexes the query tokens, and $\arg\max_i$ denotes the maximum across the feature dimension. However, this importance score is intrinsic to $Q_l$ itself and does not reflect how $Q_l$ is influenced by the attention mechanism. Explaining the attention matrix is a key component of post-hoc methods for interpreting transformer models (Wu et al., 2024a). In the case of cross-attention, the query and key matrices originate from different inputs, and thus the resulting attention matrix is not necessarily square. To better understand how $Q_l$ contributes within the attention process, we define its attention-conditioned importance scores as $\mathbf{S}(Q_l; A_l)$, the query importance modulated by the attention matrix $A_l$. We approximate this as:

$$\mathbf{S}(Q_l; A_l) \propto \frac{\partial L^c}{\partial A_l} \cdot Q_l,$$

where $\cdot$ is matrix product. From the previous step, we have already obtained the attention importance map $\frac{\partial L^c}{\partial A_l} \odot A_l$. We now seek a transformation that allows us to infer $\mathbf{S}(Q_l; A_l)$ from this. The attention matrix is computed via scaled dot-product as $A_l = Q_l K_l^\top$ with softmax and $\sqrt{d}$ ignored for simplicity. We can express with linear operations (e.g., ReLU, $\mathbb{E}_H$, and $(\cdot) + I$ ignored for simplicity.:

$$\mathbf{S}(A_l) = \frac{\partial L^c}{\partial A_l} \cdot Q_l K_l^\top,$$

To isolate the influence of $Q_l$, we need to eliminate $K_l^\top$ from the right-hand side. Since $K_l$ is not guaranteed to be square or invertible, we employ the Moore-Penrose pseudoinverse:

$$\frac{\partial L^c}{\partial A_l} \cdot Q_l K_l^\top = \mathbf{S}(A_l)$$
$$\frac{\partial L^c}{\partial A_l} \cdot Q_l = \mathbf{S}(A_l) \cdot K_l (K_l^\top K_l)^{-1} \in \mathbb{R}^{N \times d}.$$

This yields a decomposition of attention importance into the query space. Then, the importance scores corresponding to the token part can be obtained following:

$$\omega_l^A = \arg\max_i \left(\frac{\partial L^c}{\partial A_l} \cdot Q_l\right)_{i,j} \in \mathbb{R}^N,$$

where $i$ indexes the feature dimension, $j$ indexes the query tokens, and $\arg\max_i$ denotes maximum taken over feature dimension. However, to ensure robustness, particularly in cases where $Q_l$ is also influenced by other layers. We conservatively combine this result with the intrinsic query importance:

$$\mathbf{S}(Q_l; A_l) = \max\left(\omega_l^A, \omega_l^Q\right).$$

Here, the maximum is applied element-wise to capture the strongest importance attribution from either source.

### 3.2 Quantifying Key Importance from Cross-Attention

Similar to the approach used to extract query importance scores, the key matrix importance can also be quantified into two components: (1) the intrinsic importance of the key matrix, denoted as

$\mathbf{S}(K_l)$, and (2) the attention-conditioned importance, $\mathbf{S}(K_l; A_l)$, which reflects how the key matrix contributes to the attention mechanism.

The intrinsic importance of the key matrix with respect to the loss $L^c$ for class $c$ can be estimated using a GradCAM-style formulation:

$$\mathbf{S}(K_l) = \text{ReLU} \left( \frac{\partial L^c}{\partial K_l} \odot K_l \right) .$$

To obtain token-level importance scores from this matrix, we compute the column-wise maximum:

$$\omega_l^K = \arg \max_i \mathbf{S}(K_l)_{i,j} \in \mathbb{R}^{N'} .$$

where $i$ indexes the feature dimension, $j$ indexes the key tokens, and $\arg \max_i$ denotes the maximum across the feature dimension ($d$). Similar to the issue encountered in query attention quantification, the attention matrix is no longer necessarily square for key attention quantification. However, compared to decomposing query importance, extracting key importance from the attention matrix is more straightforward, since attention explicitly maps queries into the key space. Thus, we can directly analyze the attention matrix to determine which key tokens exert the strongest influence on the query representations. Because transformer models rely primarily on token-level outputs, we focus on interpreting token-level activations. The attention matrix $A \in \mathbb{R}^{N \times N'}$ indicates how each query token (rows) attends to the key tokens (columns). To evaluate the overall importance of each key token in guiding the query representations, we compute the maximum relevance of each key across all queries and attention heads:

$$\omega_l^{A'} = \arg \max_i \left( \mathbb{E}_H \left( \text{ReLU} \left( \frac{\partial L^c}{\partial A_{i,j}} \cdot A_{i,j} \right) \right) \right) \in \mathbb{R}^{N'},$$

where $\mathbb{E}_H$ denotes averaging over attention heads and $i$ and $j$ index the queries and keys respectively, and $\arg \max_i$ denotes the maximum across the feature dimension.

Finally, we combine this attention-derived importance with the intrinsic importance to produce a robust estimate of key token relevance:

$$\mathbf{S}(K_l; A_l) = \max \left( \omega_l^{A'}, \omega_l^K \right),$$

where the maximum is taken element-wise to reflect the highest attribution signal from either source.

### 3.3 Aggregation of Layer Importance Scores

Inspired by the attention flow perspective (Abnar & Zuidema, 2020), we aggregate token-level importance scores across layers to track how relevance propagates from the final output back through the decoder and encoder layers. Let $k$ denote the index of the first decoder layer (with cross-attention) encountered when traversing the model from the output layer backwards. All subsequent layers with smaller indices are assumed to be encoder layers with self-attention. To capture how importance propagates through these layers, we define the aggregated token-level importance scores at layer $k$, denoted by $\tilde{\mathbf{S}}_k$, recursively as follows:

$$\tilde{\mathbf{S}}_k = \begin{cases} \mathbf{S}(Q_k; A_k) \cdot \tilde{\mathbf{S}}_{k+1} & \text{(query)} \\ \mathbf{S}(K_k; A_k) \cdot \tilde{\mathbf{S}}_{k+1} & \text{(key)} \end{cases} .$$

In models with multiple decoder blocks that contain cross-attention, importance interactions may diverge and converge at different points. To handle such cases, we adopt a conservative strategy and aggregate importance via element-wise maximum to retain the most influential attribution signal:

$$\tilde{\mathbf{S}}_k = \begin{cases} \max \left( \mathbf{S}(Q_k; A_k), \tilde{\mathbf{S}}_{k+1} \right) & \text{(query)} \\ \max \left( \mathbf{S}(K_k; A_k), \tilde{\mathbf{S}}_{k+1} \right) & \text{(key)} \end{cases} .$$

These recursive rules ensure that attention importance is correctly traced through both cross-attention and self-attention components. Consequently, if the explanation path contains any decoder block with cross-attention, the final output of our QCAI method will be a vector of token-level importance scores, indicating the contribution of each input token to the model's prediction.

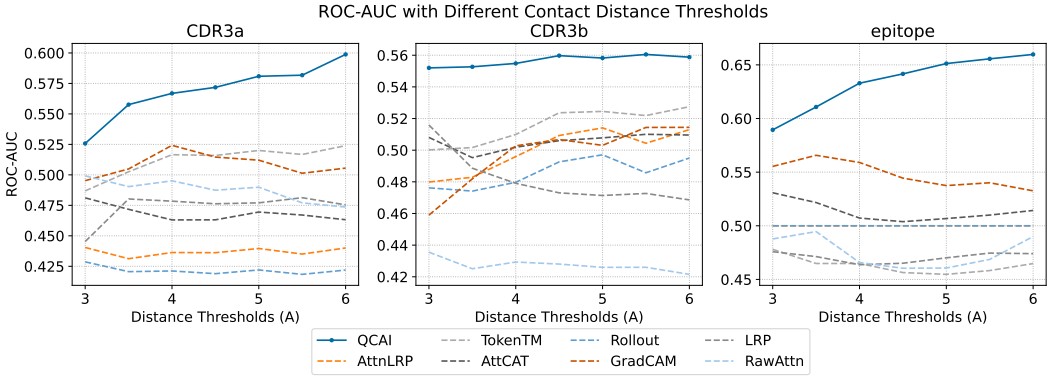

Figure 2: ROC-AUC of predicted importance scores for TCR-pMHC binding site identification across a threshold of interaction distances demonstrates that QCAI surpasses competing methods in all cases.

## 4  EXPERIMENTAL RESULTS AND DISCUSSION

We first evaluate our proposed QCAI method using a state-of-art BERT-based model named TULIP, a transformer architecture tailored for predicting TCR-pMHC binding, which focuses on the role of cross-attention and outperforms one of the widely used baseline models, NetTCR-2.2 (Jensen & Nielsen, 2024). TULIP adopts an encoder-decoder design and processes three modalities in parallel: CDR3a, CDR3b, and peptide sequences (Meynard-Piganeau et al., 2024). Each encoder independently transforms input sequences into latent feature representations, while decoder layers model inter-sequence interactions (Devlin et al., 2019; Vaswani et al., 2017). As a self-regressive generative model, TULIP estimates the conditional probability distribution of each sequence (e.g., peptide) conditioned on the others (e.g., CDR3a, CDR3b) (Meynard-Piganeau et al., 2024).

We compare our QCAI method against several existing post-hoc interpretability techniques, including AttnLRP (Achtibat et al., 2024), TokenTM (Wu et al., 2024a), AttCAT (Qiang et al., 2022), Rollout (Abnar & Zuidema, 2020), GradCAM (Selvaraju et al., 2017), LRP (Binder et al., 2016), and RawAttn (Wiegreffe & Pinter, 2019). For methods that require aggregation across all layers and compute on attention matrix, we apply them exclusively to the self-attention layers and omit cross-attention components, as these competing methods do not support cross-attention explanations. All experiments were implemented in Python using the PyTorch framework. Testing was conducted on a local workstation equipped with two NVIDIA A2000 GPUs, 16 Intel Xeon E5 CPU cores, and 64 GB of RAM.

### 4.1  A BENCHMARK FOR TCR-pMHC BINDING INTERPRETATION

To quantitatively assess the quality of interpretability methods, we constructed a benchmark that we call TCR-XAI using structural data from TCR-pMHC complexes. We collected 274 valid samples from the STCRDab (Leem et al., 2018) and TCR3d 2.0 (Lin et al., 2025) datasets, which consist of 213 (77.7%) MHC-I samples and 61 (22.3%) MHC-II samples. Only samples with complete TCR $\alpha$ and $\beta$ chains, full peptide sequences, intact CDR3 regions, and non-overlapping MHC and peptide chain IDs were retained. Statistics of the benchmark set are discussed in Section A.9 of the Appendix. For each sample, we computed residue-level distances: (1) from each CDR residue to the closest atom in the peptide, and (2) from each peptide residue to the closest atom in any CDR region. A smaller distance indicates a stronger interaction, and atomic distance as a proxy for ground-truth importance. We believe this is a simple, yet highly useful assumption since protein folding and, by extension, protein-protein interactions are most routinely evaluated by the closeness of packing, which signals the exclusion of water molecules and demands the formation of all possible hydrogen bonds (without water molecules). Other types of interatomic interactions such as hydrophobic contacts and ionic bonds contribute to binding, but they generally cannot be realized without exclusion of water. Thus, the formation of a stable protein-protein interface has a sharp distance threshold, above-which the interaction is not likely to be stable. The importance of residue-level distance is evident in prior work, starting with TCRdist Mayer-Blackwell et al. (2021),

a classic unsupervised method for TCR-pMHC binding prediction. It defines the TCRdist distance as "the similarity-weighted mismatch distance between the potential pMHC-contacting loops of the two receptors." Using distance as an indicator is also common in TCR-pMHC transformer model explanations, though typically for qualitative rather than quantitative evaluation (e.g., PISTE (Feng et al., 2024) and TCR-BERT (Wu et al., 2024b)).

## 4.2 ROC ANALYSIS AND PERTURBATION EXPERIMENTS

To evaluate the explainability of different post-hoc interpretation methods, we quantitatively assess their ability to identify true TCR-pMHC binding sites using the TCR-XAI benchmark. We computed ROC curves (ROC curves for each threshold are shown in Section A.7 in the Appendix.) by comparing predicted residue importance against ground-truth binding site annotations derived from structural data, where the ground-truth was defined according to distance threshold between 3 and 6 Å, with the ROC using predicted importance as the threshold. As shown in Figure 2, QCAI achieves AUCs of 0.5492, 0.5489, and 0.6024 for CDR3a, CDR3b, and peptide respectively and consistently outperforms other methods. Notably, QCAI exceeds 0.6 on the peptide chain, demonstrating strong alignment between its predicted importance scores and the underlying structural binding interactions.

We also conducted perturbation studies on to assess whether each method identifies important residues; we adopt two commonly used metrics: Log-Odds Score (LOdds) and Area Over the Perturbation Curve (AOPC). AOPC measures explanation quality by averaging the drop in model confidence as top-k important features are removed. Higher AOPC indicates better alignment between explanation and model behavior. LOdds computes the change in log-odds of the model's prediction before and after perturbing a feature. A larger LOdds value indicates greater importance of the perturbed feature. Perturbation is implemented by replacing the $k$ highest-scoring tokens with padding tokens ($< PAD >$). We evaluate the CDR3a, CDR3b and peptide chains separately, with $k=4$ for the CDR3a and CDR3b chains, and $k=7$ for peptides to match the average number of predicted binding residues across TCR-XAI.

| Chains | CDR3a $_{k=4}$ | | CDR3b $_{k=4}$ | | Peptide $_{k=7}$ | |
|---|---|---|---|---|---|---|
| | LOdds | AOPC | LOdds | AOPC | LOdds | AOPC |
| **QCAI (Ours)** | **-3.328** | 0.014 | **-3.498** | **0.045** | **-1.470** | **0.013** |
| AttnLRP (Achtibat et al., 2024) | -2.481 | 0.020 | -2.662 | 0.032 | -0.017 | 0.000 |
| TokenTM (Wu et al., 2024a) | -2.195 | 0.021 | -2.383 | 0.032 | -0.736 | 0.012 |
| AttCAT (Qiang et al., 2022) | -2.825 | 0.020 | -3.131 | 0.044 | -0.694 | 0.006 |
| Rollout (Abnar & Zuidema, 2020) | -2.356 | **0.022** | -2.653 | 0.032 | -0.044 | -0.001 |
| GradCAM (Selvaraju et al., 2017) | -2.700 | 0.019 | -3.112 | 0.038 | -1.004 | 0.009 |
| LRP (Binder et al., 2016) | -2.938 | 0.020 | -3.127 | 0.043 | -1.167 | 0.011 |
| RawAttn (Wiegreffe & Pinter, 2019) | -2.734 | 0.015 | -3.250 | 0.039 | -0.691 | 0.010 |

Table 1: Perturbation experiment results using fixed thresholds. Thresholds for the $\alpha$ and $\beta$ chains are $k=4$, and for the peptide chain $k=7$. The average number of binding regions are 3.64, 4.12, and 7.05 for $\alpha$, $\beta$, and peptide chains respectively.

Table 1 shows that QCAI consistently outperforms other methods across most metrics. QCAI achieves the most negative LOdds and highest AOPC scores in the CDR3b and peptide chains, indicating greater disruption to the model's confidence when informative residues are perturbed. Although Rollout outperforms QCAI in AOPC on the CDR3a chain, QCAI still achieves the best LOdds score.

## 4.3 IDENTIFICATION OF BINDING REGION RESIDUES WITH IMPORTANCE SCORES

Using the TCR-XAI benchmark we construct an evaluation metric that we call *Binding Region Hit Rate* (BRHR). To compute BRHR, we first choose a percentile threshold $t \in (0, 1]$ and identify the top $t$ fraction of residues with respect to highest importance scores $\mathbf{S}$. Each of these residues is marked a hit if its interaction distance is in the top $t$ fraction of interaction distances. We compute the hit rate for each input sequence type in each sample and take the mean across TCR-XAI to obtain

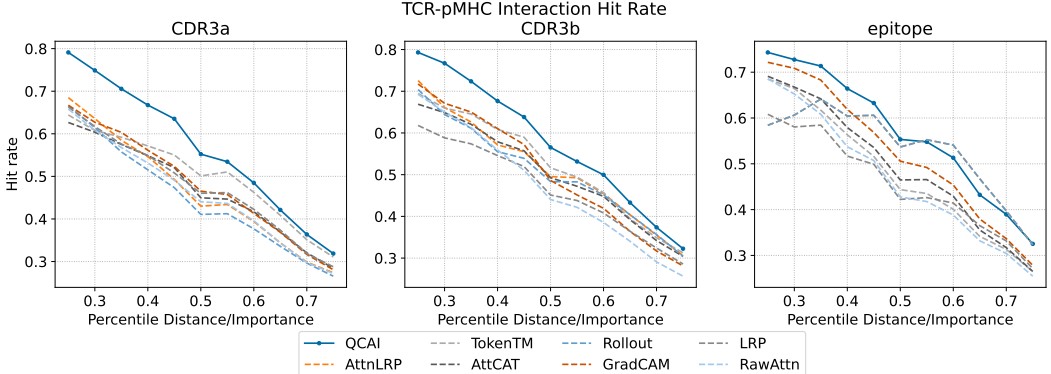

Figure 3: Comparison of TCR-pMHC Binding Region Hit Rate (BRHR) across different methods on different chains. At any selected percentile of distance/importance, the higher the hit rate the more closely the importance tracks physical interaction distance. QCAI surpasses other methods in all practical cases.

the final BRHR. This metric reflects the proportion of true binding residues (according to structural proximity) that are successfully identified by the explanation method.

As shown in Figure 3, our method achieves state-of-the-art performance compared to all other explanation methods. For the peptide chain, our method consistently outperforms all other methods before the 50th percentile. After this threshold other methods prevail but have high false positive rate of other methods (as seen in the ROC analysis). We postulate that the latter effect is due to the fact that these methods can only access self-attention weights from the encoder and cannot benefit from the regulatory influence of cross-attention layers.

## 4.4 CASE STUDIES

To highlight the ability of QCAI to assist in the interpretation of TCR-pMHC binding we discuss two specific examples, one for CD8+ T cells and one for CD4+ T cells. In both cases the analysis of importance using QCAI finds residue positions in CDR3s that form critical contacts with epitope peptides and, by revealing unconstrained positions in longer CDR loops, can explain large differences in TCR-peptide-HLA binding affinity.

In the first case study (Figure 4(a)) we consider the immunodominant CD8+ T-cell peptide from the influenza matrix protein which has been used to understand influenza T cell response. Multiple crystal structures (1OGA (Stewart-Jones et al., 2003) and 5TEZ (Yang et al., 2017)) of different TCRs recognizing this peptide have revealed a common mode of binding that involves the insertion of a single CDR3b sidechain (R98 in the 1OGA structure) into a notch between the peptide and the HLA-A2 alpha-2 helix and, otherwise, makes numerous contacts with the HLA-A2, whose shape depends on the peptide. In one distinct example, the TCR in the 5TEZ structure is rotated by 40 degrees around the HLA-TCR axis to create a very different group of TCR-HLA-A2 contacts, but this TCR also places a CDR3b sidechain (W99 in the 5TEZ structure) in the notch between peptide and HLA-A2 alpha-2 helix. Consistent with the common aspect of binding, the QCAI evaluation finds importance in the position of the notch-binding residue and in several N-terminal flanking positions of CDR3b. The distinct aspect of binding for the two TCRs arises in the longer and less-constrained CDR3a for the 5TEZ TCR, which may explain its 25-fold lower affinity than for the 1OGA TCR. We note that for both binding orientations, AttnLRP and TokenTM produce weaker importance scores overall.

The second case study considers a self-antigen in the autoimmune disease of rheumatoid arthritis. The HLA-DR4-bound citrullinated peptide, named vimentin-64cit59-71, has been analyzed in the complex with two different TCRs (Loh et al., 2024) (indicated with PDB codes, 8TRR and 8TRQ in Figure 4(b)). The QCAI evaluation finds an overall similar number of important positions in the two TCRs, including a concentration of importance along one edge of the hairpin formed by the CDR3a in both TCRs (highlighted with a dark outline in Figure 4(b)). The CDR3a contributes the largest direct contact with the peptide in both complexes. Interestingly, the CDR3b of the 5-fold-lower-

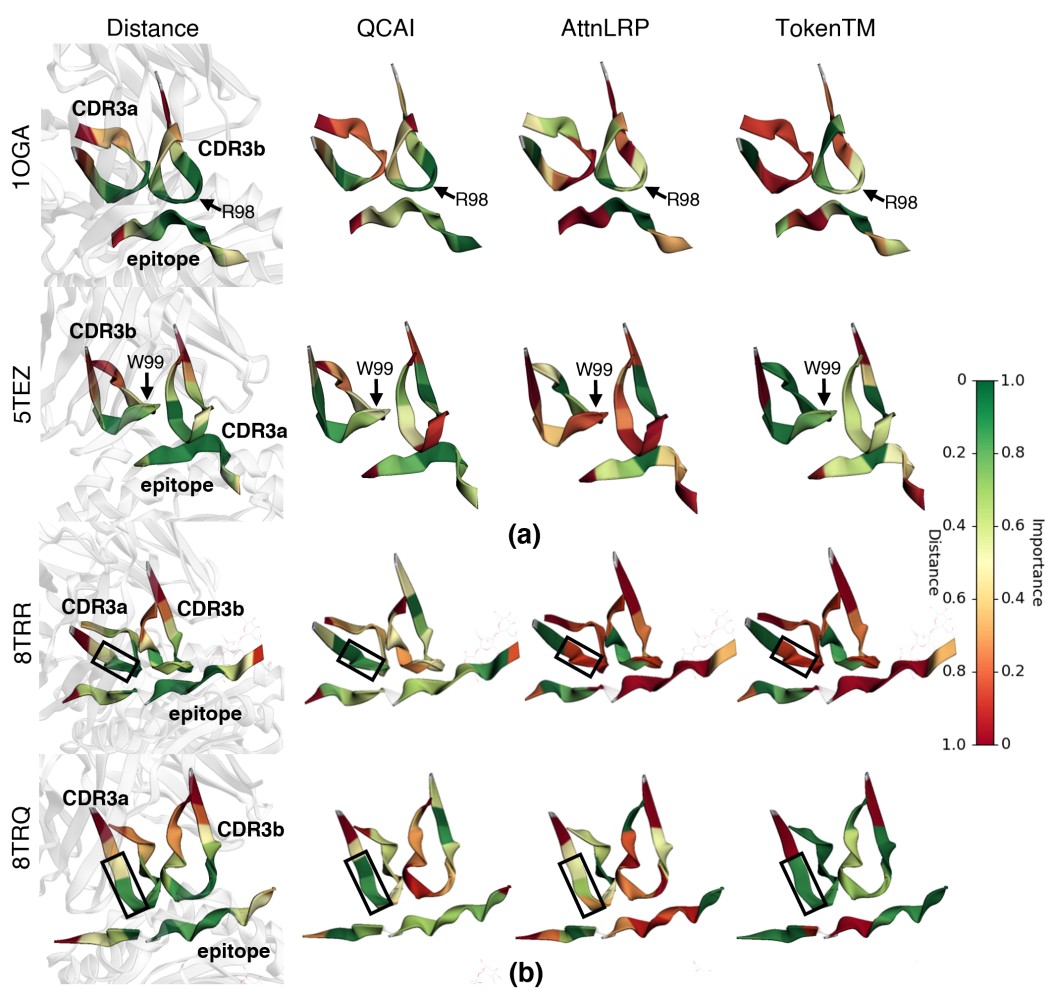

Figure 4: Case studies on systems from TCR-XAI. (a) We consider the same TCR-pMHC bound in two distinct binding orientations. For this system QCAI identifies key residues from both orientations. (b) We consider the same pMHC bound to two distinct TCRs. Here QCAI identifies the importance of the hairpin region of CDR3a in both cases.

affinity 8TRQ complex is longer and contains more positions of lower importance, again suggesting that the entropic cost of ordering this loop is responsible for the reduced affinity. For this case study, AttnLRP does not produce meaningful results while TokenTM does not capture the importance of residues in the peptide proximal to the CDR3a hairpin.

To investigate how QCAI explanations differ for similar TCR-pMHC complexes, we conducted a case study on two TCR-MHCII-peptide structures, 2PXY and 2Z31, which investigates whether a germline-encoded motif structurally guides TCR recognition of MHC (Feng et al., 2007). They differ by two amino acids in the CDR3b loop (Feng et al., 2007). To convince chain alignment, amino acids were re-indexed starting from 1 for each chain. As shown in Figure 5, QCAI with TULIP assigns similar importance scores to the peptide in both complexes but produces different pattern of importance for the CDR3b loop. Both complexes correctly highlight A5 as an important contact region, and QCAI identifies additional contact sites in 2PXY. In 2PXY, residues S6 and G7 receive higher scores, whereas the corresponding region in 2Z31 receives lower scores, where are also the contact regions. These results indicate that QCAI can detect critical contact regions even with minor sequence changes. However, such changes can affect the overall explanation quality.

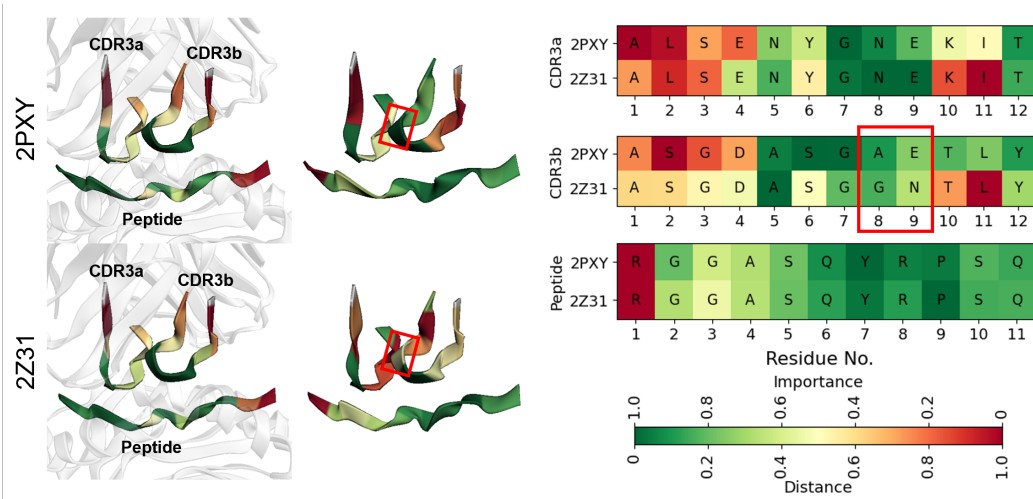

Figure 5: Case studies of two closely related TCR-pMHC complexes from TCR-XAI. These complexes differ by only two amino acids in the CDR3b, highlighted in the figure with red rectangles.

## 5 CONCLUSIONS

In this paper, we present Quantifying Cross-Attention Interaction (QCAI) to interpret the cross-attention in the decoders of transformer models, aiming to better understand encoder-decoder TCR-pMHC binding prediction models. QCAI quantifies the importance of the cross-attention matrix into contributions from query and key inputs, revealing how they influence each other. To rigorously evaluate the explanations, we created a new structural explanation benchmark, TCR-XAI, along with a novel evaluation metric, the Binding Region Hit Rate (BRHR). On this benchmark, QCAI achieves state-of-the-art results across perturbation metrics (LOdds and AOPC), ROC-AUC, ROC curve analysis, and BRHR.

### 5.1 FUTURE WORK

In future work, we plan to pursue two primary directions: (1) extending the metrics used to evaluate explainability, and (2) applying QCAI to broader range of immunological and protein-protein interaction tasks. Beyond distance-based measures, energy functions (e.g., REF15 (Alford et al., 2017)) offer a promising alternative for quantifying explanations in TCR-pMHC binding prediction. Investigating a range of energy-based models to better understand the relationship between explainability and protein energetics will be an important next step.

Given the emergence of several cross-attention models for protein-protein interactions and immunological tasks, such as PALM-H3 (He et al., 2024) for antigen generation, UniPMT (Zhao et al., 2025) for peptide-MHC prediction, ProtAttBA (Liu et al., 2025) for antibody-antigen prediction, and HB-Former (Zhang et al., 2024) for human-virus interaction identification, QCAI provides a method for opening the black box of cross-attentions in these models and revealing their underlying mechanisms. In addition, QCAI can be extended beyond these applications. For instance, we have already applied it to CLIP encoders with cross-attention, as discussed in Appendix A.12. Exploring broader applications of QCAI across these tasks and domains is an important direction for future work.

Transformers are widely used for TCR-pMHC binding prediction, but they remain black-box models. While post-hoc methods like QCAI improve explainability, they cannot directly integrate these insights into prediction. Beyond post-hoc methods, an important future direction is to develop explain-by-design models that provide inherent explainability and utilize mechanistic TCR-pMHC insights to improve predictive performance.

**Code Availability:** The source code is publicly available at our project website (https://qcai.jiarui.li/) and on GitHub (https://github.com/Jiarui0923/QCAI).

ACKNOWLEDGMENTS

The authors acknowledge support from the Harold L. and Heather E. Jurist Center of Excellence for Artificial Intelligence at Tulane University. This work was also supported by the National Institutes of Health (U54-CA260581) through the Tulane University COVID Antibody and Immunity Network (TUCAIN); Tulane SOM Pilot Funding for "MHCII Pathway Processing of SARS-CoV-2 Spike"; the Tulane Center of Excellence for Emerging and Re-emerging Infectious Diseases (CEERID) Pilot Research Program for "Large-Scale Validation of a Novel Parallel Algorithm for Computational Epitope Prediction"; and the Lavin-Bernick Faculty Grant Proposal Research and Scholarly Activities Support for "Research Trainee Support for Modeling Antigen Processing and HLA Immunopeptidomics".

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

# A   SUPPLEMENTARY MATERIAL

## A.1   POST-HOC EXPLANATION METHODS

A variety of explainable AI (XAI) methods have been developed to interpret deep learning models (Saranya & Subhashini, 2023). These methods fall into two broad categories: explain-by-design, which integrates interpretability into the model architecture (Dwivedi et al., 2023), and post-hoc, which analyzes model behavior after training (Kenny et al., 2021). Post-hoc approaches offer a promising avenue for interpreting TCR-pMHC models and uncovering the underlying factors driving binding predictions. Several families of post-hoc methods have been proposed, including:

- The Class Activation Map (CAM) (e.g., CAM (Zhou et al., 2016), GradCAM (Selvaraju et al., 2017), GradCAM++ (Chattopadhay et al., 2018))
- Layer-wise Relevance Propagation (LRP) (e.g., LRP (Binder et al., 2016), Partial LRP (Voita et al., 2019), Conservative LRP (Ali et al., 2022), AttnLRP (Achtibat et al., 2024))
- Attention-based methods (e.g., Raw Attention (Wiegreffe & Pinter, 2019), Attention Rollout (Abnar & Zuidema, 2020), AttCAT (Qiang et al., 2022))
- Model-specific hybrid methods (e.g., TokenTM (Wu et al., 2024a), GAE (Chefer et al., 2021))

These techniques have been successfully applied to TCR-pMHC models. For example, TEPCAM interpret a attention-CNN with attention map (Chen et al., 2024), while TCR-BERT relies on attention weight analysis for interpretability (Wu et al., 2024b). These efforts have revealed structural determinants of TCR-pMHC binding. However, existing post-hoc methods primarily support encoder-only or co-attention mechanisms (Chefer et al., 2021), limiting their applicability to modern encoder-decoder models, which consists of cross-attention. This poses a major barrier to understanding how such models capture TCR-pMHC interactions.

## A.2   CLASS ACTIVATION MAPS

Class Activation Map (CAM)-based methods have achieved significant success in explaining Convolutional Neural Networks (CNNs) by generating class-discriminative localization maps. Grad-CAM (Selvaraju et al., 2017), one of the most effective CAM methods, leverages the gradient of the class score $L^c$ with respect to the feature maps $F_d$ from the last convolutional layer. These gradients are used to compute importance weights for each feature map channel, enabling spatial localization of the regions most relevant for class $c$. The importance weight $w_d^c$ for feature map $F_d$ is computed as:

$$w_d^c = \mathbb{E}\left(\frac{\partial L^c}{\partial F_d}\right),$$

where $\mathbb{E}$ denotes global average and $w_d^c$ represents the global average pooled gradient for feature map $F_d$. The final CAM is then computed as a weighted sum over channels, followed by a ReLU activation:

$$\text{GradCAM}^c = \text{ReLU}\left(\sum_d w_d^c F_d\right).$$

The resulting heatmap is upsampled to the input resolution to highlight input regions most relevant to the prediction for class $c$.

### A.2.1   ATTENTION ROLLOUT

CAM-based approaches are primarily designed for CNNs. To interpret transformer-based models, Attention Rollout was proposed by Abnar & Zuidema (2020), which estimates the flow of attention across layers. This method computes how information propagates through the self-attention mechanism across layers. Given the raw attention weights $W_l^A$ for layer $l$, the augmented attention matrix is defined as

$$A_l = \frac{1}{2}(W_l^A + I),$$

where $I$ is the identity matrix, modeling the residual connection. The cumulative attention, or rollout, is then computed recursively:

$$R_l = \begin{cases} A_l R_{l-1}, & \text{if } l > 0 \\ A_l, & \text{if } l = 0 \end{cases},$$

capturing the total attention contribution from input tokens through layer $l$.

## A.3 TCR-PMHC BINDING PREDICTION

T cells are important component of our immune system, which can be mainly catogorized in two CD8+ and CD4+ T cells. CD8+ T cells are initiated through the Major Histocompatibility Complex I (MHCI) pathway, while CD4+ T cells are initiated through the MHCII pathway. Epitope prediction for CD8+ T cells has had remarkable success, while the mechanisms of CD4+ T cell response are less understood. T cell immune response can be viewed as consisting of two stages of recognition. In the first stage, a antigen is taken up by antigen-presenting cells (APCs), where it undergoes joint processing (i.e., cleavage) and binding to Major Histocompatibility Complex II (MHCII) molecules. Peptide-MHC complexes are then presented on the APC cell surface (Davis & Bjorkman, 1988; Neefjes et al., 2011). In the second stage, T cell receptors (TCRs) on T cells "recognize" pMHC complexes and a T cell response is initiated. TCR recognition is mediated by its $\alpha$ and $\beta$ domains, which consist of variable (V), joining (J), constant (C), and, in the $\beta$ chain, diversity (D) regions (Bosselut, 2019).

Accurate prediction of T cell responses requires a comprehensive understanding of both of these stages (Peters et al., 2020; Nielsen et al., 2020). Early efforts in the area of computational epitope prediction focused on characterizing peptide-MHCII binding using allele-specific machine learning models (Nielsen et al., 2020) with tools such as SMM (Peters & Sette, 2005; Kim et al., 2009), NetMHC (Lundegaard et al., 2008; Nielsen et al., 2003), NetMHCpan (Hoof et al., 2009; Nielsen et al., 2007), and NetMHCcons (Karosiene et al., 2012). More recent work has focused on modeling antigen processing computationally with the Antigen Processing Likelihood (APL) algorithm (Mettu et al., 2016; Bhattacharya et al., 2023; Li et al., 2024a;b; Charles et al., 2022), which seeks to model the contributions of antigen structure on which peptides are made available for MHCII binding.

Accurately predicting TCR-pMHC binding remains critical for advancing quantitative immunology and adaptive immunity research (Hudson et al., 2023). For this stage of prediction, both unsupervised and supervised methods have been developed (Hudson et al., 2023; 2024). Unsupervised methods process cluster TCR sequencing datasets through dimensionality reduction and clustering (Dash et al., 2017; Glanville et al., 2017) through a carefully chosen similarity metric (e.g., TCRdist3 (Mayer-Blackwell et al., 2021)). These methods cluster TCRs by analyzing their complementarity-determining regions (CDRs) using only TCR sequence data, without requiring binding labels or epitope information (e.g., GIANA (Zhang et al., 2021), ClusTCR (Valkiers et al., 2021), GLIPH2 (Huang et al., 2020) iSMART (Zhang et al., 2020)). The resulting cluster labels serve as the output for each input TCR sequence (Hudson et al., 2024) and are typically analyzed by practitioners to guide and supplement experimental methods. In contrast, supervised machine learning techniques make use of large amounts of TCR-pMHC data for training (Hudson et al., 2023) from databases such as VDJdb (Bagaev et al., 2020), McPAS-TCR (Tickotsky et al., 2017) and the IEDB (Vita et al., 2019). Supervised approaches (e.g. TITAN (Weber et al., 2021), STAPLER (Kwee et al., 2023), ERGO2 (Springer et al., 2021), MixTCRpred (Croce et al., 2024), NetTCR2.2 (Jensen & Nielsen, 2024), TULIP (Meynard-Piganeau et al., 2024)) use a variety of deep learning models providing reasonable performance and generalization capability.

### A.3.1 TCR-PMHC BINDING PROBLEM FORMATION

The TCR-pMHC binding prediction problem can be formulated as a classification task: given the TCR alpha ($\alpha$) and beta ($\beta$) chains, an epitope $e$, and an MHC molecule $m$, the model predicts whether the pair binds (binder) or does not bind (non-binder). The TCR chains and the epitope are proteins or peptides, typically represented as amino acid sequences. Formally, we define amino acid units as $a \in \mathbb{A}$, where $\mathbb{A}$ is the set of amino acid characters. For a single TCR-pMHC binding case, $\alpha = [a_i^\alpha]_{i=1}^{N_\alpha}$, $\beta = [a_i^\beta]_{i=1}^{N_\beta}$, and $e = [a_i^e]_{i=1}^{N_e}$, with $N_\alpha, N_\beta, N_e \in \mathbb{Z}^+$ representing the sequence lengths. The MHC allele type is denoted by $m \in M$, where $M$ is the set of all MHC alleles. The

pMHC-TCR binding classification is formulated as a conditional probability: $p_{\text{bind}} = P(e|\alpha, \beta, m)$. If $p_{\text{bind}} > t$, where $t \in [0, 1]$, the case is classified as positive, otherwise negative.

### A.4 TCR-pMHC Prediction Transformer Models

Transformers (Vaswani et al., 2017), as a successful deep learning models in different areas, have a series of variants such as Bidirectional Encoder Representations from Transformers (BERT) (Devlin et al., 2019) and Generative Pre-training Transformers (GPT) (Radford et al., 2018). These models support multi-sequence inputs and excel in modeling interactions, are well-suited for this task. Because TCR-pMHC interactions are determined by interactions among the TCR $\alpha$ and $\beta$ chains, epitope, and MHC, several state-of-the-art models, such as TULIP (Meynard-Piganeau et al., 2024) and cross-TCR-interpreter (Koyama et al., 2023), adopt encoder-decoder transformer architectures to learn these complex relationships.

**TULIP:** TULIP is a transformer-based model with an encoder-decoder architecture designed for TCR-pMHC binding prediction. It operates through three parallel modality processing pipelines, processing CDR3a, CDR3b, and epitope sequences separately (Meynard-Piganeau et al., 2024). The encoders transform the input sequences into feature representations, while the decoders model interactions across different sequences (Devlin et al., 2019; Vaswani et al., 2017). As an auto-regressive generative model,

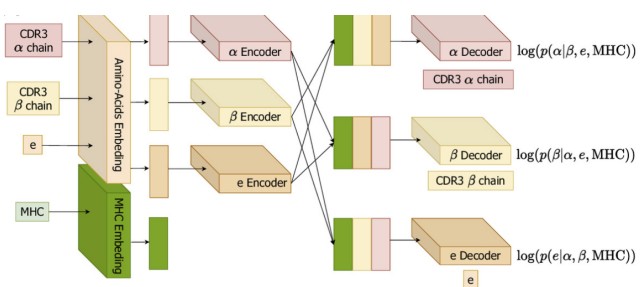

Figure 6: The architecture figure of TULIP model (Meynard-Piganeau et al., 2024).

TULIP computes the conditional probability distribution of sequences (e.g., epitope) given others (e.g., CDR3a, CDR3b, and MHC) during training (Meynard-Piganeau et al., 2024). For evaluation, TULIP retains only the epitope stream to produce the binding score. In this setting, the peptide features serve as the query, while the CDR3a and CDR3b features are used as the keys and values in the cross-attention module. To compute gradients for TULIP, we design an amino-acid-wise loss function. The ground truth is derived from the TCR alpha, TCR beta, and epitope sequences. These sequences are first one-hot encoded, and the model's predicted probabilities are compared against them using a negative log-likelihood (NLL) loss. This formulation allows us to attribute importance scores at the amino acid level based on how well the model reconstructs each residue.

**CrossTCRInterpreter:**
CrossTCRInterpreter is an encoder-decoder transformer for TCR-pMHC binding prediction (Koyama et al., 2023). It takes the CDR regions of the alpha and beta chains, along with the peptide sequence, as inputs. The CDR alpha and beta chains are concatenated using a colon (:) to form the TCR input. The TCR and peptide sequences are then independently encoded by an encoder module. Subsequently, cross-attention is employed to model the interaction between the two inputs and predict whether the pair represents a binder or a non-binder. We apply a binary classification loss to extract the model gradients.

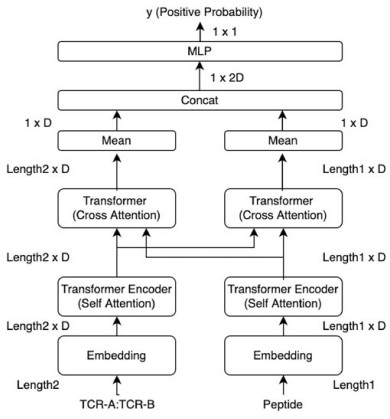

Figure 7: The architecture of CrossTCRInterpreter model (Koyama et al., 2023).

**BERTrand:** BERTrand is an encoder-only transformer model (Myronov et al., 2023). It takes the TCR beta chain and peptide sequence as inputs. These two sequences are concatenated using a `<SEP>` token and are processed jointly by a transformer encoder as an integrated input. Similar to CrossTCRInterpreter, BERTrand is a classification model designed to predict whether the TCR-pMHC pair is a binder or a non-binder. We apply a binary classification loss to obtain the model gradients.

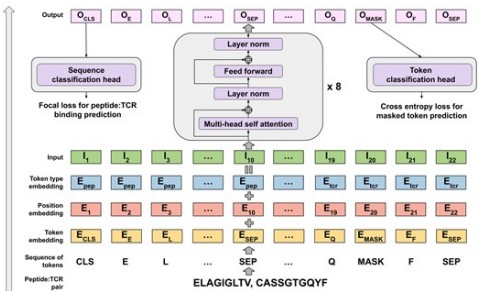

Figure 8: The architecture figure of BERTrand model (Myronov et al., 2023).

### A.5 PERTURBATION EXPERIMENTS

We evaluated the robustness of interpretability methods using perturbation-based metrics across varying values of $k$. Figure 9 presents the comparison results for both AOPC and LOdds across all chains.

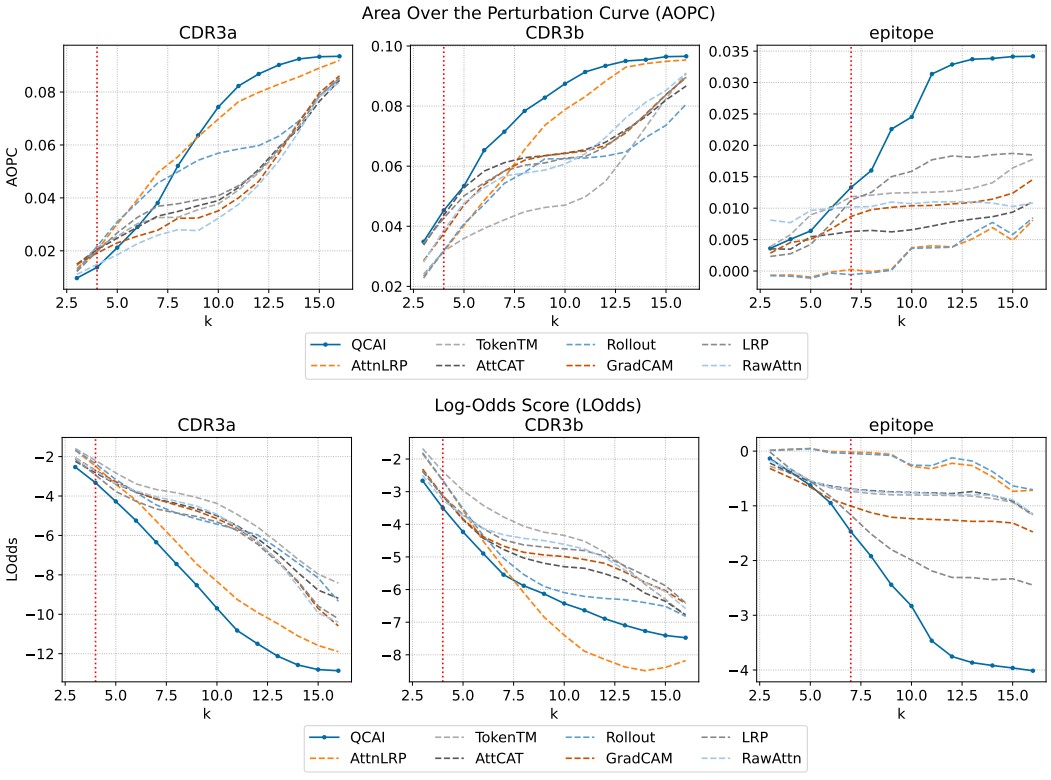

Figure 9: Comparison of Area Over the Perturbation Curve (AOPC) and Comparison of Log-Odds Score (LOdds) across different values of $k$ for all chains.

We also conducted with an integrated dataset that includes data from VDJdb, IEDB and McPAS-TCR. For both AOPCs and LOdds, the thresholds for peptide, CDR3a, and CDR3b are 7, 4, and 4 respectively.

| | | AOPCs | | LOdds | |
|---|---|---|---|---|---|
| Chain | Method | TCR-XAI | Integrated | TCR-XAI | Integrated |
| peptide | QCAI | 0.014 | **0.036** | -1.62 | **-0.84** |
| peptide | TokenTM | 0.013 | 0.026 | -0.77 | 0.09 |
| peptide | AttnLRP | 0.012 | 0.026 | -0.42 | -0.50 |
| CDR3a | QCAI | 0.014 | 0.020 | -3.50 | **-2.63** |
| CDR3a | TokenTM | 0.021 | 0.020 | -2.43 | -2.35 |
| CDR3a | AttnLRP | 0.020 | 0.020 | -2.72 | -2.44 |
| CDR3b | QCAI | 0.048 | **0.027** | -3.61 | **-3.08** |
| CDR3b | TokenTM | 0.033 | 0.025 | -2.53 | -2.78 |
| CDR3b | AttnLRP | 0.034 | 0.024 | -2.82 | -2.88 |

Table 2: AOPCs and LOdds comparison on TCR-XAI and Integrated datasets.

## A.6 MAXIMUM VS. AVERAGE FOR AGGREGATION

High attention weights indicate meaningful interactions and so we used maximum across different cross-attention layers to retain all activated signals. Ablation studies in the table below show that max generally outperforms average, with small exceptions on peptide BRHR and CDR LOdds.

| Chain | Mix | ROC-AUC(3.4) | BRHR.25 | AOPCs | LOdds |
|---|---|---|---|---|---|
| peptide | Max. | 0.60 | 74.3 | **0.014** | **-1.52** |
| peptide | Avg. | 0.60 | **76.7** | 0.013 | -1.51 |
| CDR3a | Max. | **0.55** | **79.1** | **0.014** | -3.37 |
| CDR3a | Avg. | 0.50 | 72.6 | 0.013 | **-3.51** |
| CDR3b | Max. | **0.55** | **79.3** | **0.046** | -3.54 |
| CDR3b | Avg. | 0.54 | 75.3 | 0.045 | **-3.63** |

Table 3: Maximum vs. Average for aggregation comparison across chains for ROC-AUC (3.4), BRHR, AOPCs, and LOdds.

## A.7 ROC CURVES OF RESIDUE-LEVEL IMPORTANCE SCORES FOR BINDING REGION IDENTIFICATION

We compared QCAI with other methods using ROC curves. We mainly set the distance threshold at 3, 3.4, 4, and 5 Å. 3.4 Å is chosen because it corresponds to the van der Waals diameter between two carbon atoms, implying that residues within this distance are considered to be in contact.

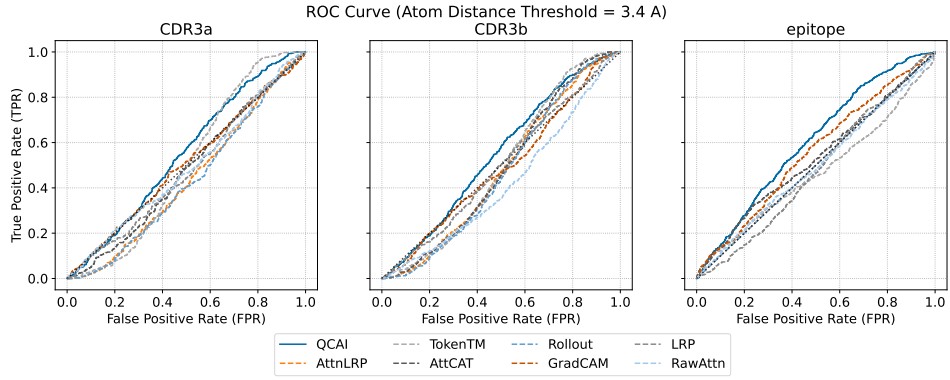

Figure 10: ROC curve comparison of the alpha, beta, and epitope chains between QCAI and other post-hoc methods. The distance threshold is set to 3.4 Å.

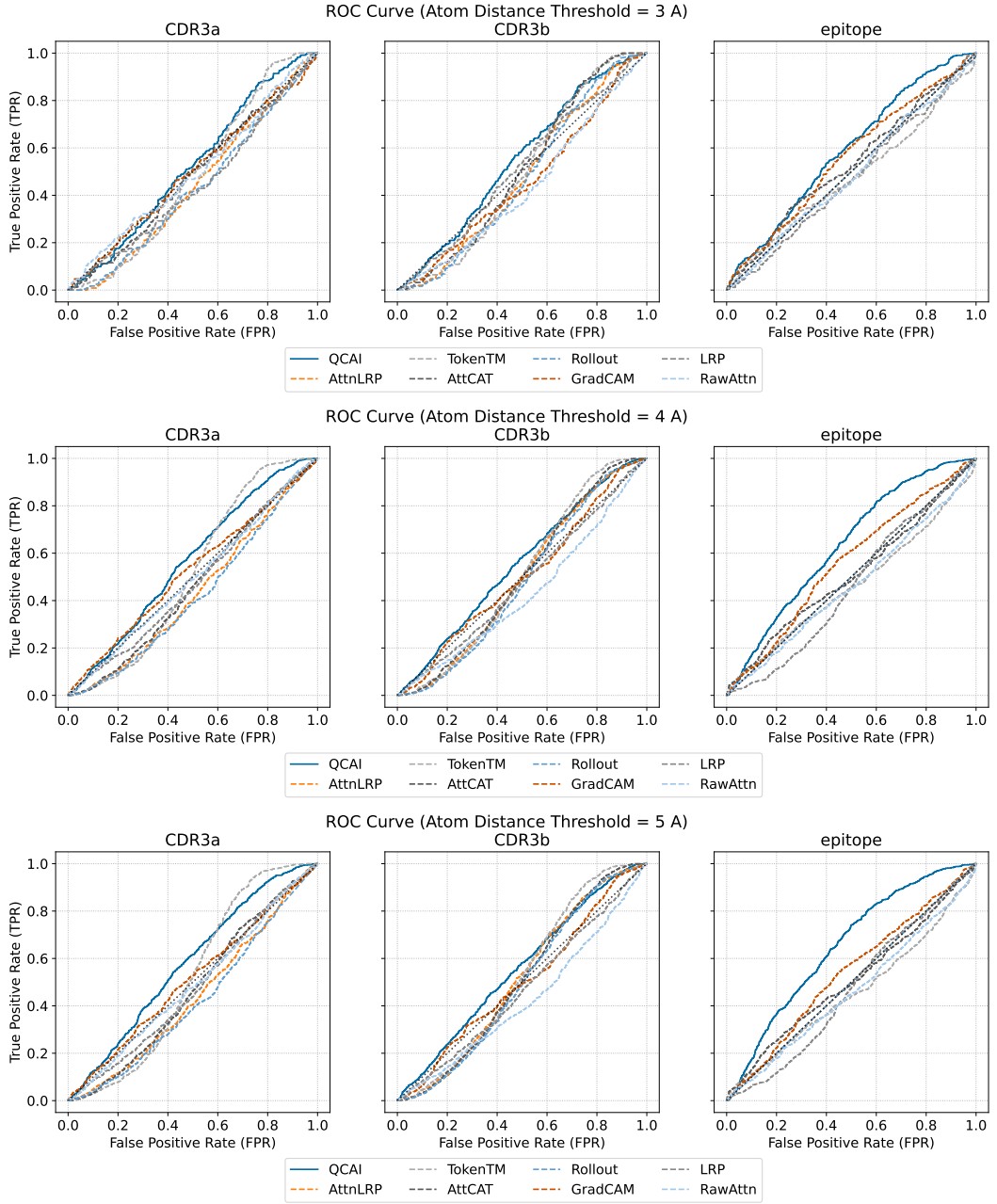

Figure 11: ROC curve comparison of the alpha, beta, and epitope chains between QCAI and other post-hoc methods. The distance thresholds are set to 3, 4, and 5 Å.

## A.8 BINDING REGION HIT RATE

We compare the Binding Region Hit Rate (BRHR) across the TCR $\alpha$ and $\beta$ chains as well as the epitope region for different explanation methods. Here, HR.$t$ denotes the hit rate calculated based on the top $t$ percentile of importance scores.

| Chain | Method | HR.25 | HR.30 | HR.40 | HR.50 |
|---|---|---|---|---|---|
| epitope | **QCAI (Ours)** | **74.3(±24.5)**% | **72.7(±24.6)**% | **66.4(±19.8)**% | **55.3(±15.7)**% |
| | AttnLRP (Achtibat et al., 2024) | 58.4(±29.2)% | 60.6(±25.9)% | 60.4(±19.8)% | 53.7(±15.8)% |
| | TokenTM (Wu et al., 2024a) | 68.5(±29.6)% | 66.4(±29.4)% | 56.3(±25.9)% | 44.4(±20.7)% |
| | AttCAT (Qiang et al., 2022) | 69.1(±30.8)% | 66.8(±30.0)% | 57.9(±23.9)% | 46.5(±18.4)% |
| | Rollout (Abnar & Zuidema, 2020) | 58.4(±29.2)% | 60.6(±25.9)% | 60.4(±19.8)% | 53.7(±15.8)% |
| | GradCAM (Selvaraju et al., 2017) | 72.2(±29.5)% | 70.9(±29.3)% | 61.9(±24.9)% | 50.6(±19.9)% |
| | LRP (Binder et al., 2016) | 60.8(±31.4)% | 58.0(±29.8)% | 51.7(±21.4)% | 42.2(±17.3)% |
| | RawAttn (Wiegreffe & Pinter, 2019) | 68.6(±27.7)% | 65.2(±26.0)% | 53.7(±23.5)% | 42.8(±20.0)% |
| CDR3a | **QCAI (Ours)** | **79.1(±20.1)**% | **74.9(±19.4)**% | **66.7(±16.7)**% | **55.2(±16.6)**% |
| | AttnLRP (Achtibat et al., 2024) | 68.5(±24.5)% | 63.6(±25.1)% | 54.6(±20.9)% | 43.0(±19.4)% |
| | TokenTM (Wu et al., 2024a) | 64.4(±30.8)% | 60.8(±30.2)% | 57.1(±27.1)% | 50.1(±20.9)% |
| | AttCATT (Qiang et al., 2022) | 62.7(±25.0)% | 60.4(±24.9)% | 54.9(±23.4)% | 45.0(±19.9)% |
| | Rollout (Abnar & Zuidema, 2020) | 66.6(±25.2)% | 61.5(±24.7)% | 51.5(±20.7)% | 41.1(±18.8)% |
| | GradCAM (Selvaraju et al., 2017) | 66.7(±26.7)% | 62.7(±25.5)% | 56.1(±20.1)% | 46.5(±17.0)% |
| | LRP (Binder et al., 2016) | 66.3(±27.1)% | 61.7(±26.6)% | 54.9(±21.8)% | 46.0(±19.0)% |
| | RawAttn (Wiegreffe & Pinter, 2019) | 65.8(±27.2)% | 60.8(±25.3)% | 53.1(±22.5)% | 44.0(±17.1)% |
| CDR3b | **QCAI (Ours)** | **79.3(±19.0)**% | **76.7(±18.9)**% | **67.7(±16.2)**% | **56.5(±14.9)**% |
| | AttnLRP (Achtibat et al., 2024) | 72.6(±25.3)% | 66.1(±23.4)% | 57.1(±21.8)% | 49.5(±17.8)% |
| | TokenTM (Wu et al., 2024a) | 69.5(±27.5)% | 66.0(±26.7)% | 60.8(±22.0)% | 51.6(±18.3)% |
| | AttCAT (Qiang et al., 2022) | 66.9(±25.9)% | 64.9(±24.4)% | 57.9(±23.2)% | 49.3(±19.4)% |
| | Rollout (Abnar & Zuidema, 2020) | 70.4(±25.6)% | 64.4(±23.3)% | 55.5(±21.5)% | 48.3(±18.0)% |
| | GradCAM (Selvaraju et al., 2017) | 71.7(±26.8)% | 67.1(±27.1)% | 61.0(±24.3)% | 48.6(±19.1)% |
| | LRP (Binder et al., 2016) | 61.8(±26.3)% | 58.8(±23.1)% | 54.6(±20.6)% | 45.1(±18.5)% |
| | RawAttn (Wiegreffe & Pinter, 2019) | 69.3(±24.3)% | 65.0(±22.2)% | 55.9(±19.8)% | 44.0(±17.8)% |

Table 4: The Binding Region Hit Rate comparison among TCR alpha and beta chains and epitope between various methods. The HR.$t$ denotes the hit rate computed based on the top $t$ percentile.

To consider performance relative to training set similarity we consider the change in BRHR of samples indexed by the Levenshtein distance each input modality to the TULIP training dataset (this approach is also used in the original TULIP paper). In the Table A.8 each cell represents the BRHR of all samples with the minimum Levenshtein distance to the sequences of TULIP training dataset smaller than the given threshold. As a sample's distance between the sequences of TCR-XAI and the sequences of training dataset increases we find that the BRHR score of QCAI on TULIP decreases slightly but remains reliable with the BRHR drop within 0.05, which is small relative to the BRHR difference between QCAI and other methods. This shows that QCAI's performance is preserved even as samples differ from the training set.

| Levenshtein Distance ($d$) | $1 > d$ | $2 > d$ | $3 > d$ | $4 > d$ | $5 > d$ | $6 > d$ | $7 \leq d$ |
|---|---|---|---|---|---|---|---|
| Peptide | .77(.23) | .77(.24) | .80(.23) | .77(.24) | .75(.25) | .74(.25) | .76(.25) |
| CDR3a | .83(.17) | .81(.18) | .84(.19) | .82(.19) | .79(.20) | .79(.20) | .79(.20) |
| CDR3b | .83(.20) | .80(.21) | .77(.21) | .78(.21) | .78(.20) | .78(.20) | .78(.20) |

Table 5: The change in BRHR for samples indexed by their Levenshtein distance.

To examine how model confidence and prediction outcomes affect BRHR, we further compare BRHR on positive and negative samples for both TULIP and Cross-TCR-Interpreter. Because TULIP provides only relative binding likelihood scores, we perform QCAI analysis on Cross-TCR-Interpreter separately for its predicted positive and negative samples. For TULIP, we additionally set a manual threshold by treating the top 50% scoring pairs as positive and the remaining 50% as negative. The BRHR results shown in Table A.8 indicate that negative samples decreased the quality of explanation comparing to the positive samples.

| BRHR | Cross-TCR-Interpreter | Cross-TCR-Interpreter | TULIP | TULIP |
|---|---|---|---|---|
| **Samples** | Positive | Negative | Positive | Negative |
| Peptide | .61 | .55 | .75 | .74 |
| CDR3a | .41 | .46 | .79 | .79 |
| CDR3b | .87 | .87 | .81 | .79 |

Table 6: The BRHR of predicted positive and negative samples.

## A.9 TCR-XAI BENCHMARK

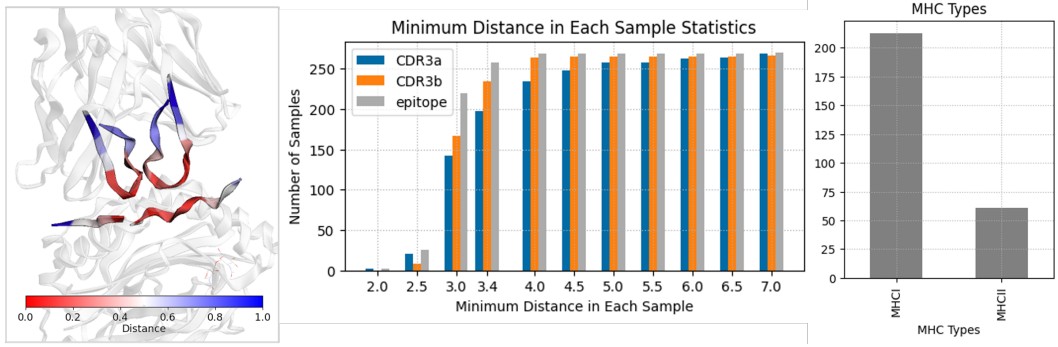

Figure 12: In the example (8TRQ) from the TCR-XAI benchmark, the peptide, CDR3a, and CDR3b regions are highlighted based on their residue-level distances to the nearest interacting residues. Additionally, we report statistics for the minimum distance in each sample and MHC distribution.

We have compiled 274 samples from the STCRDab (Leem et al., 2018) and TCR3d 2.0 (Lin et al., 2025) datasets. Only samples with fully provided CDR3 regions and peptide sequences were selected. Among them, 213 (77.7%) are MHC-I and 61 (22.3%) are MHC-II complexes. For each

sample, we computed the distance from each residue in the CDR3 regions to the nearest atom in the peptide, and vice versa from the peptide residues to the CDR3 regions. The resulting dataset includes both the CDR3 and peptide sequences along with their corresponding residue-level distances. Since the model lacks structural input, we allow a one-residue positional tolerance to account for minor attention shifts. To this end, we smooth each method's output importance scores by convolving them with the kernel $[1/3, 1/3, 1/3]$ prior to evaluation. The detailed information can be found in Table 9. Compared with the TULIP training dataset, there are 176 distinct epitopes, and none appears in more than 3.3% (9) of the samples.

## A.10    COMPUTATIONAL EFFICIENCY OF QCAI

We evaluate QCAI efficiency based on datasets including VDJdb, IEDB, McPAS-TCR, and TCR-XAI. All evaluations are conducted on CPU (32 E5 cores). QCAI involves pseudo-inverse operations making it more computationally expensive than alternative approaches, but it is still relatively efficient on a per sample basis. For example benchmark sets with thousands of test samples would need on the order of seconds for QCAI evaluation - this is far smaller than what would be needed by practitioners.

| Method | TCR-XAI | McPAS-TCR | VDJdb | IEDB |
|---|---|---|---|---|
| QCAI | 2.19 ms | 2.18 ms | 1.30 ms | 1.90 ms |
| TokenTM | 0.11 ms | 0.15 ms | 0.05 ms | 0.11 ms |
| AttnLRP | 0.04 ms | 0.02 ms | 0.02 ms | 0.04 ms |

Table 7: Milliseconds per sample for each method across different datasets.

## A.11    ABLATION STUDY: QCAI ON CROSS- VS. SELF-ATTENTION

To investigate whether QCAI applied to cross-attention or self-attention contributes more to the final explanation, we compare QCAI applied only to cross-attention, only to self-attention, and to both, using perturbation experiments. Applying QCAI solely to self-attention is equivalent to Rollout. As shown in Figure A.11 and Table A.11, the performance of QCAI on cross-attention alone is comparable to applying it to both cross- and self-attention, and both outperform Rollout. These results indicate that cross-attention is the main contributor to the final explanation and plays a significant role in cross-attention incorporated transformers.

| | CDR3a$_{k=4}$ | | CDR3b$_{k=4}$ | | Peptide$_{k=7}$ | |
|---|---|---|---|---|---|---|
| | LOdds | AOPC | LOdds | AOPC | LOdds | AOPC |
| QCAI | -3.328 | 0.014 | -3.498 | 0.045 | -1.470 | 0.013 |
| QCAI (Cross-Attention) | -3.728 | 0.017 | -3.511 | 0.048 | -1.417 | 0.012 |
| Rollout (Self-Attention) | -2.356 | 0.022 | -2.653 | 0.032 | -0.044 | -0.001 |
| AttnLRP | -2.481 | 0.020 | -2.662 | 0.032 | -0.017 | 0.000 |
| TokenTM | -2.195 | 0.021 | -2.383 | 0.032 | -0.736 | 0.012 |

Table 8: Comparison of AOPC and LOdds for QCAI applied to cross-attention only, self-attention only (Rollout), and both.

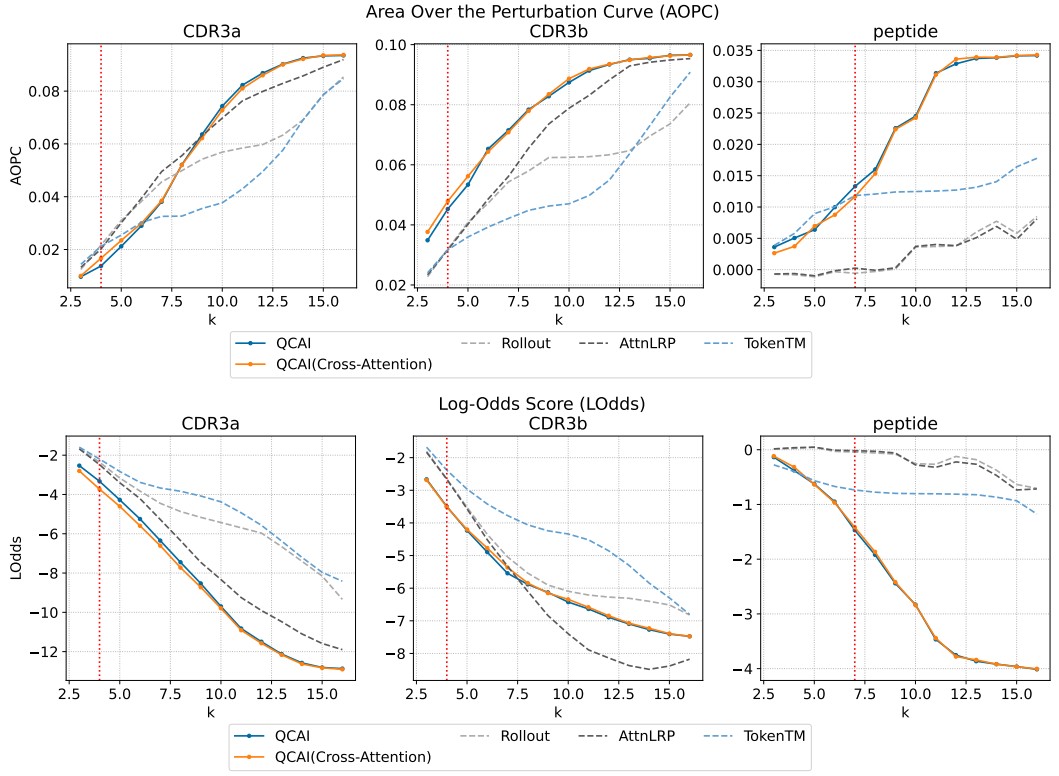

Figure 13: Comparison of AOPC and LOdds for QCAI applied to cross-attention only, self-attention only (Rollout), and both.

## A.12 APPLICATION OF VISION-LANGUAGE MODELS

Since QCAI can be applied broadly to cross-attention modules, we illustrate its use in a vision-language model (VLM). For this case study, we employ CLIP Radford et al. (2021), a widely used vision foundation model. CLIP provides separate vision and text encoders with aligned features, so we added a cross-attention layer to fuse image features (as key and value) with text features (as query). We use a subset of the MS-COCO dataset Lin et al. (2014), containing 73,000 images for multi-label classification. The dataset is split 9:1, with 65,700 training samples and 7,300 test samples. The input consists of an image–text pair, where the text is generated following the CLIP paper's recommendation as "a photo of a ..." with the corresponding labels (e.g., "a photo of a cat", "a photo of a couch") Radford et al. (2018). Features extracted via cross-attention are used for label prediction. After 100 epochs of training, the model achieves 94.08% accuracy and 0.9997 ROC-AUC on the test set.

QCAI is then applied to analyze the model. Since each image-text pair has multiple labels, gradients and QCAI are computed for one label at a time. Figure A.12 presents case studies on both training and test samples, demonstrating that QCAI can identify interactions between the two input modalities in cross-attention and highlight the relative importance of the image and text for a given classification label.

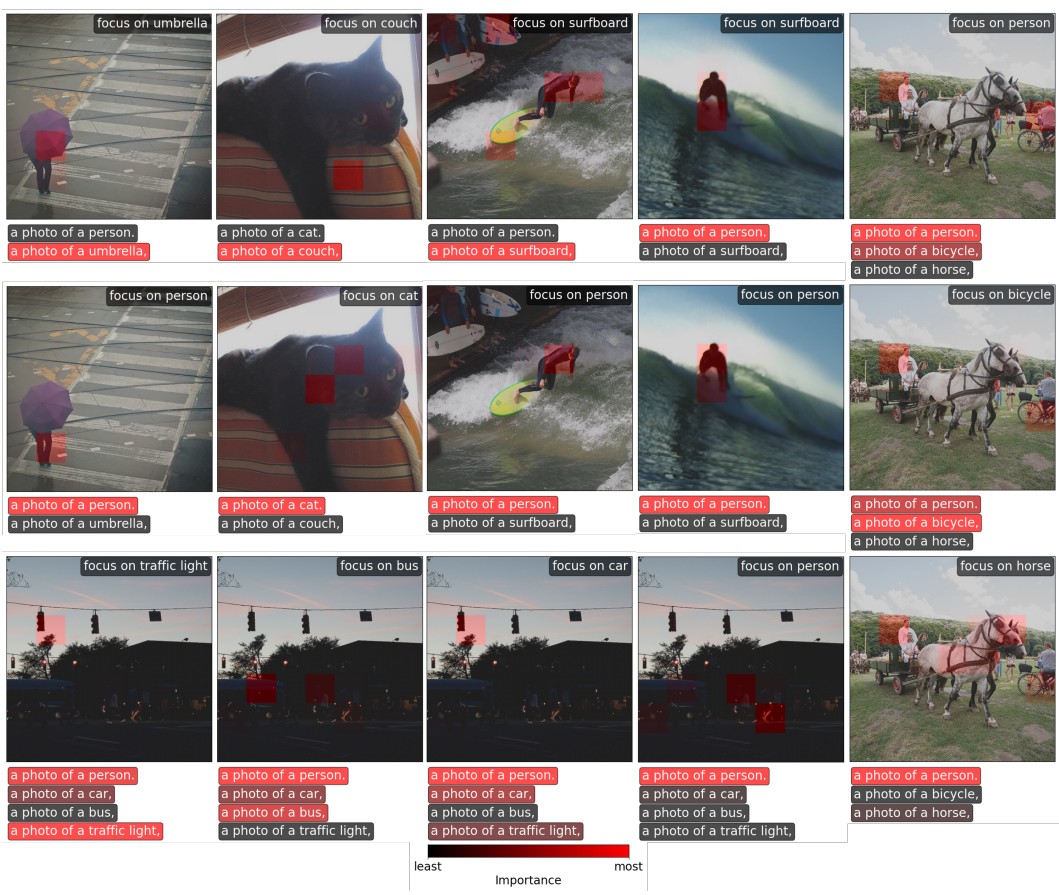

Figure 14: Example of QCAI explaining CLIP with cross-attention.

## A.13 TCR-XAI BENCHMARK SAMPLES

| PDB | MHC | Peptide | CDRA3 | CDRB3 |
|-----|-----|---------|-------|-------|
| 8TRQ | MHCII | GVYATSSAVRLR | ALGDHSGSWQLI | ASSLRTGANSDYT |
| 4OZI | MHCII | QPFPQPELPYP | LVGDGGSFSGGYNKLI | SAGVGGQETQY |
| 2AK4 | MHCII | LPEPLPQGQLTAY | ALSGFYNTDKLI | ASPGLAGEYEQY |
| 5EU6 | MHCI | YLEPGPVTV | AVLSSGGSNYKLT | ASSFIGGTDTQY |
| 7PBE | MHCI | YLQPRTFLL | VVNINTDKLI | ASSSANSGELF |
| 4Z7W | MHCII | PSGEGSFQPSQENPQ | AVGETGANNLF | ASSEARRYNEQF |
| 6V18 | MHCII | GGYRAPAKAAAT | ALSDSGSFNKLT | ASSLDWGGQNTLY |
| 8EO8 | MHCI | LPFDKATIM | AADGGAGSYQLT | SAGPTSGRTDTQY |
| 5WKH | MHCI | GTSGSPIINR | GLGDAGNMLT | ASSLGQGLLYGYT |
| 7T2B | MHCII | ATGLAWEWWRTVYE | ATDKKGGATNKLI | ASSQGGGEQY |
| 7SG1 | MHCII | QPFPQPELPYGSGGS | LVGGLARDMR | SVALGSDTGELF |
| 3W0W | MHCI | RFPLTFGWCF | GTYNQGGKLI | ASSGASHEQY |
| 5W1V | MHCI | VMAPRTLIL | AGQPLGGSNYKLT | ASSANPGDSSNEKLF |
| 6AVF | MHCI | APRGPHGGAASGL | LVGEILDNFNKFY | ASSQRQEGDTQY |
| 5NHT | MHCI | ELAGIGILTV | AVGGGADGLT | ASSQGLAGAGELF |
| 3TPU | MHCI | FLSPFWFDI | AVSAKGTGSKLS | ASSDAPGQLY |
| 2P5E | MHCI | SLLLMWIITQC | AVRPLLDGTYIPT | ASSYLGNTGELLF |
| 4P2O | MHCII | PADPLAFFSSAIKGGGGSLV | AALRATGGNNKLT | ASSLNWSQDTQY |
| 7NME | MHCI | QLPRLFPLL | AEPSGNTGKLI | ASSLHHEQY |
| 5JZI | MHCI | KLVALGINAV | AYGEDDKII | ASRRGPYEQY |
| 8I5C | MHCI | VVGAVGVGK | AARDSNYQLI | ASGDTGGYEQY |
| 4E41 | MHCII | GELIGILNAAKVPAD | AVDRGSTLGRLY | ASSQIRETQY |
| 6AM5 | MHCI | SMLGIGIVPV | AVNFGGGKLI | ASSLSFGTEAF |
| 3E2H | MHCI | QLSPFPFDL | AVSLERPYLT | ASGGGGTLY |
| 3MV8 | MHCI | HPVGEADYFEY | AVQDLGTSGSRLT | ASSARSGELF |
| 3KPR | MHCI | EEYLKAWTF | ILPLAGGTSYGKLT | ASSLGQAYEQY |
| 5HYJ | MHCI | AQWGPDPAAA | AMRGDSSYKLI | ASSLWEKLAKNIQY |
| 5KS9 | MHCII | APSGEGSFQPSQENPQ | AVALNNNAGNMLT | ASSVAPGSDTQY |
| 7T2D | MHCII | ATGLAWEWWRTVYE | ALSGSARQLT | ASSHREGETQY |
| 1KJ2 | MHCI | KVITFIDL | AARYQGGRALI | TCSAAPDWGASAETLY |
| 6RPB | MHCI | SLLMWITQV | AVKSGGSYIPT | ASSYLNRDSALD |
| 6UON | MHCI | GADGVGKSAL | AAAMDSSYKLI | ASSDPGTEAF |
| 1ZGL | MHCII | VHFFKNIVTPRTPG | ALSGGDSSYKLI | ASSLADRVNTEAF |
| 3PQY | MHCI | SSLENFRAYV | ILSGGSNYKLT | ASSFGREQY |
| 4Y19 | MHCII | QPLALEGSLQKRG | AASVYAGGTSYGKLT | ASRPRRDNEQF |
| 6AVG | MHCI | APRGPHGGAASGL | LVVDQKLV | ASSGGHTGSNEQF |
| 6V15 | MHCII | GGYAPAKAAAT | ALSPSNTNKVV | ASSLDWGVNTLY |
| 4Z7U | MHCII | APSGEGSFQPSQENPQ | ILRDRSNQFY | ASSTTPGTGTETQY |
| 7RM4 | MHCI | HMTEVVRHC | ALDIYPHDMR | ASSLDPGDTGELF |
| 4QOK | MHCI | EAAGIGILTV | AVNVAGKST | AWSETGLGTGELF |
| 3PWP | MHCI | LGYGFVNYI | AVTTDSWGKLQ | ASRPGLAGGRPEQY |
| 7N6E | MHCI | YLQPRTFLL | VVNRNNDMR | AGQVTNTGELF |
| 8WUL | MHCI | VVGAVGVGK | AARSSGSWQLI | ASSQDRGDSAHTLY |
| 3DXA | MHCI | EENLLDFVRF | IVWGGYQKVT | ASRYRDDSYNEQF |
| 6RP9 | MHCI | SLLMWITQV | ALTRGPGNQFY | ASSSPGGVSTEAF |
| 4MJI | MHCI | TAFTIPSI | ATDDDSARQLT | ASSLTGGGELF |
| 6V19 | MHCII | GGYAPAKAAAT | ALSDSSSFSKLV | ASSLDWASQNTLY |
| 7RDV | MHCII | EGRVRVNSAYQS | AASDDNNNRIF | ASGGQSNERLF |
| 3E3Q | MHCI | QLSPFPFDL | AVSDPPPLLT | ASGGGGTLY |
| 4MS8 | MHCI | SPAEEAGFFL | AVSAKGTGSKLS | ASSDAPGQLY |
| 2F53 | MHCI | SLLMWITQC | AVRPTSGGSYIPT | ASSYVGNTGELF |
| 3QDJ | MHCI | AAGIGILTV | AVNFGGGKLI | ASSLSFGTEAF |
| 6BJ2 | MHCI | IPLTEEAEL | ALSHNSGGSNYKLT | ASSFRGGKTQY |
| 3QDM | MHCI | ELAGIGILTV | AGGTGNQFY | AISEVGVGQPQH |
| 5TIL | MHCI | KAPYNFATM | AALYGNEKIT | ASSDAGGRNTLY |
| 6VMX | MHCI | RPPIFIRRL | AFGSSNTGKLI | ASSQDLFTGGYT |

Table 9: The samples contained in TCRxAI benchmarks

| PDB | MHC | Peptide | CDRA3 | CDRB3 |
|-----|-----|---------|-------|-------|
| 7RTR | MHCI | YLQPRTFLL | AVNRDDKII | ASSPDIEQY |
| 8EN8 | MHCI | LPFDKSTIM | AADGGAGSYQLT | SAGPTSGRTDTQY |
| 1YMM | MHCII | ENPVVHFFKNIVTP | ATDTTSGTYKYI | SARDLTSGANNEQF |
| 5C07 | MHCI | YQFGPDFPIA | AMRGDSSSYKLI | ASSLWEKLAKNIQY |
| 3VXU | MHCI | RFPLTFGWCF | GTYNQGGKLI | ASSGASHEQY |
| 6VQO | MHCI | HMTEVVRHC | AMSGLKEDSSYKLI | ASSIQQGADTQY |
| 1J8H | MHCII | PKYVKQNTLKLAT | AVSESPFGNEKLT | ASSSTGLPYGYT |
| 8ENH | MHCI | LPFEKSTIM | AADGGAGSYQLT | SAGPTSGRTDTQY |
| 2P5W | MHCI | SLLMWITQC | AVRPLLDGTYIPT | ASSYLGNTGELF |
| 3UTT | MHCI | ALWGPDPAAA | AMRGDSSYKLI | ASSLWEKLAKNIQY |
| 7Q99 | MHCI | NLSALGIFST | AVNVAGKST | AWSETGLGTGELF |
| 6ZKZ | MHCI | RLPAKAPL | AVTNQAGTALI | ASSYSIRGSRGEQF |
| 4GG6 | MHCII | SGEGSFQPSQENP | ILRDGRGGADGLT | ASSVAVSAGTYEQY |
| 6TMO | MHCI | EAAGIGILTV | AVNDGGRLT | AWSETGLGMGGWQ |
| 3QIU | MHCII | ADLIAYLKQATKG | AAEPSSGQKLV | ASSLNNANSDYT |
| 5WKF | MHCI | GTSGSPIVNR | GLGDAGNMLT | ASSLGQGLLYGYT |
| 8SHI | MHCI | VRSRRLRL | ATDALYSGGGADGLT | ASSYSEGEDEAF |
| 5D2N | MHCI | NLVPMVATV | ILDNNNDMR | ASSLAPGTTNEKLF |
| 5KSA | MHCII | QPQQSFPEQEA | AVQFMDSNYQLI | ASSVAGTPSYEQY |
| 6MTM | MHCI | FEDLRVLSF | GTERSGGYQKVT | ASSMSAMGTEAF |
| 2BNQ | MHCI | SLLMWITQV | AVRPTSGGSYIPT | ASSYVGNTGELF |
| 4Z7V | MHCII | SGEGSFQPSQENP | ILRDSRAQKLV | ASSAGTSGEYEQY |
| 2F54 | MHCI | SLLMWITQC | AVRPTSGGSYIPT | ASSYVGNTGELF |
| 5BS0 | MHCI | ESDPIVAQY | AVRPGGAGPFFVV | ASSFNMATGQY |
| 6CQR | MHCII | RFYKTLRAEQASQ | AFKAAGNKLT | ASSRLAGGMDEQF |
| 5M00 | MHCI | KAVANFATM | AALYGNEKIT | ASSDDAAGGGGRNTLY |
| 7N2Q | MHCI | LRVMMLAPF | AVSNFNKFY | ASSVATYSTDTQY |
| 6EQB | MHCI | AAAAGGIIGGIILTV | AVNDGGRLT | AWSETGLGMGGWQQ |
| 4P2R | MHCII | ANGVAFFLTPFKA | AAEASNTNKVV | ASSLNNANSDYT |
| 4P2Q | MHCII | ADGLAYFRSSFKGG | AAEASNTNKVV | ASSLNNANSDYT |
| 8DNT | MHCI | LLLDRLNQL | AVREGAQKLV | ASSLDLGADEQF |
| 5E6I | MHCI | GILGFVFTL | AGPGGSSNTGKLI | ASSLIYPGELF |
| 5TJE | MHCI | KAVYNFATM | AALYGNEKIT | ASSDAGGRNTLY |
| 2J8U | MHCI | ALWGFFPVL | ALFLASSSFSKLV | ASSDWVSYEQY |
| 1LP9 | MHCI | ALWGFFPVL | ALFLASSSFSKLV | ASSDWVSYEQY |
| 3KPS | MHCI | EEYLQAFTY | ILPLAGGTSYGKLT | ASSLGQAYEQY |
| 2BNR | MHCI | SLLMWITQC | AVRPTSGGSYIPT | ASSYVGNTGELF |
| 5W1W | MHCI | VMAPRTLVL | AGQPLGGSNYKLT | ASSANPGDSSNEKLF |
| 6CQL | MHCII | RFYKTLRAEQASQ | AFKAAGNKLT | ASSRLAGGMDEQF |
| 5C09 | MHCI | YLGGPDFPTI | AMRGDSSYKLI | ASSLWEKLAKNIQY |
| 4MXQ | MHCI | SPAPRPLDL | AVSAKGTGSKLS | ASSDAPGQLY |
| 3SJV | MHCI | FLRGRAYGL | VVRAGKLI | ASGQGNFDIQY |
| 1QRN | MHCI | LLFGYAVYV | AVTTDSWGKLQ | ASRPGLAGGRPEQY |
| 3KXF | MHCI | LPEPLPQGQLTAY | ALSGFYNTDKLI | ASPGLAGEYEQY |
| 5C0A | MHCI | MVWGPDPLYV | AMRGDSSYKLI | ASSLWEKLAKNIQY |
| 7N2N | MHCI | TRLALIAPK | AVLSPVQETSGSRLT | ASSVGLFSTDTQY |
| 8ES9 | MHCI | GVYDGREHTV | AVQPLNAGNNRKLI | SAREWGGTEAF |
| 2NX5 | MHCI | EPLPGGQLTAY | AVQASGGSYIPT | ATGTGDSNPQH |
| 1G6R | MHCI | SIYRYYGL | AVSGFASALT | ASGGGGTLY |
| 8GVB | MHCI | RYPLTFGW | AVGFTGGGNKLT | ASSDRDRVPETQY |
| 8TRL | MHCII | EIFDSGNPTGEV | IVNPANTGNQFY | ASRRDYFSYEQY |
| 5D2L | MHCI | NLVPMVATV | AFITGNQFY | ASSQTQLWETQY |
| 5WLG | MHCI | SQLLNAKYL | ATVYAQGLT | ASSDWGDTGQLY |
| 5NMG | MHCI | SLFNTIAVL | AVRTNSGYALN | ASSDTVSYEQY |
| 7DZM | MHCI | TPQDLNTML | IVRGLNNAGNMLT | ASSLGIDAIY |
| 7BYD | MHCI | GGAI | LVGGGGYVLT | ASSQDLGAGEVYEQY |
| 5HHO | MHCI | GILEFVFTL | AGAGSQGNLI | ASSIRSSYEQY |

Table 10: The samples contained in TCRxAI benchmarks (continue table 1)

| PDB | MHC | Peptide | CDRA3 | CDRB3 |
| --- | --- | --- | --- | --- |
| 1QSE | MHCI | LLFGYPRYV | AVTTDSWGKLQ | ASRPGLAGGRPEQY |
| 3RGV | MHCI | WIYVYRPMGCGGS | AANSGTYQR | ASGDFWGDTLY |
| 2E7L | MHCI | QLSPFPFDL | AVSHQGRYLT | ASGGGGTLY |
| 3MBE | MHCII | GAMKRHGLDNYRGYSLG | AAEDGGSGNKLI | ASSWDRAGNTLY |
| 5M01 | MHCI | KAPANFATM | AALYGNEKIT | ASSDDAAGGGGRRNTLY |
| 5SWZ | MHCI | ASNENMETM | AASETSGSWQLI | ASSRDLGRDTQY |
| 5NMF | MHCI | SLYNTIATL | AVRTNSGYALN | ASSDTVSYEQY |
| 8GVI | MHCI | RYPLTFGW | AVVFTGGGNKLT | ASSLRDRVPETQY |
| 7N5C | MHCI | SSLCNFRAYV | ILSGGCCNYKLT | ASSFGREQY |
| 8TRR | MHCII | GVYATSSAVRLR | ALGDTGNYKYV | ASSAVNSGNTLY |
| 4OZG | MHCII | APQPELPYPQPG | IVLGGADGLT | ASSFRFTDTQY |
| 4OZH | MHCII | APQPELPYPQPGS | IVWGGATNKLI | ASSVRSTDTQY |
| 2OL3 | MHCI | SQYYYNSL | AMRGDYGGSGNKLI | TCSADRVGNTLY |
| 7QPJ | MHCI | GLYDGMEHL | AVRGTGRRALT | ASSFATEAF |
| 3VXM | MHCI | RFPLTFGWCF | AVGAPSGAGSYQLT | ASSPTSGIYEQY |
| 6EQA | MHCI | AAAAGGIIGGIILTV | AVNVAGKST | AWSETGLGTGELF |
| 2YPL | MHCI | KAFSPEVIPMF | AVSGGYQKVT | ASTGSYGYT |
| 7RK7 | MHCI | YMDGTMSQV | LVALNYGGSQGNLI | AISPTEEGGLIFPGNTIY |
| 8WTE | MHCI | VVGAVGVGK | AARSSGSWQLI | ASSQDRGDSAETLY |
| 3UTS | MHCI | ALWGPDPAAA | AMRGDSSYKLI | ASSLWEKLAKNIQY |
| 1QSF | MHCI | LLFGYPVAV | AVTTDSWGKLQ | ASRPGLAGGRPEQY |
| 1OGA | MHCI | GILGFVFTL | AGAGSQGNLI | ASSSRSSYEQY |
| 2GJ6 | MHCI | LLFGKPVYV | AVTTDSWGKLQ | ASRPGLAGGRPEQY |
| 3QDG | MHCI | ELAGIGILTV | AVNFGGGKLI | ASSLSFGTEAF |
| 2VLR | MHCI | GILGFVFTL | AGAGSQGNLI | ASSSRASYEQY |
| 7NA5 | MHCI | YGFRNVVHI | AVSNYNVLY | ASSQEPGGYAEQF |
| 8CX4 | MHCI | LRVMMLAPF | AVNSPGSGAGSYQLT | ASSVGTYSTDTQY |
| 4PRI | MHCI | HPVGEADYFEY | AVQDLGTSGSRLT | ASSARSGELF |
| 8YE4 | MHCI | NYNYLYRLF | VVNAHSGAGSYQLT | ASSETGGYEQY |
| 5M02 | MHCI | KAPFNFATM | AALYGNEKIT | ASSDAGGRNTLY |
| 2CKB | MHCI | EQYKFYSV | AVSGFASALT | ASGGGGTLY |
| 3TFK | MHCI | QLSDVPMDL | AVSAKGTGSKLS | ASSDAPGQLY |
| 7N2S | MHCI | TRLALIAPK | AVSLGTGAGSYQLT | ASSVGLYSTDTQY |
| 5KSB | MHCII | GPQQSFPEQEA | AVQASGGSYIPT | ASSNRGLGTDTQY |
| 2UWE | MHCI | ALWGFFPVL | ALFLASSSFSKLV | ASSDWVSYEQY |
| 7Q9A | MHCI | LLLGIGILVL | AVNVAGKST | AWSETGLGTGELF |
| 5C08 | MHCI | RQWGPDPAAV | AMRGDSSYKLI | ASSLWEKLAKNIQY |
| 3HG1 | MHCI | ELAGIGILTV | AVNVAGKST | AWSETGLGTGELF |
| 8I5D | MHCI | VVGAVGVGK | AASSGSWQLI | ASSLEGTVEETLY |
| 5JHD | MHCI | GILGFVFTL | AWGVNAGGTSYGKLT | ASSIGVYGYT |
| 7JWJ | MHCI | ASNENMETM | AAVTGNTGKLI | ASSRGTIHSNTEVF |
| 4MNQ | MHCI | ILAKFLHWL | AVDSATALPYGYI | ASSYQGTEAF |
| 6PY2 | MHCII | APFSEQEQPVLG | ASPQGGSEKLV | ASSSGGWGGGTEAF |
| 7DZN | MHCI | TPQDLNTML | IVRGLNNAGNMLT | ASSLGIDAIY |
| 4EUP | MHCI | ALGIGILTV | AVSGGGADGLT | ASSFLGTGVEQY |
| 7N1E | MHCI | RLQSLQTYV | ALSGFNNAGNMLT | ASSLGGAGGADTQY |
| 3QEQ | MHCI | AAGIGILTV | AGGTGNQFY | AISEVGVGQPQH |
| 2IAN | MHCII | GELIGTLNAAKVPAD | AALIQGAQKLV | ASTYHGTGY |
| 2VLJ | MHCI | GILGFVFTL | AGAGSQGNLI | ASSSRSSYEQY |
| 6CQN | MHCII | RFYKTLRAEQASQ | AFKAAGNKLT | ASSGLAGGMDEQF |
| 3VXR | MHCI | RYPLTFGWCF | AVRMDSSYKLI | ASSSWDTGELF |
| 7NMG | MHCI | LWMRLLPLL | AEPSGNTGKLI | ASSLHHEQY |
| 3D3V | MHCI | LLFGPVYV | AVTTDSWGKLQ | ASRPGLAGGRPEQY |
| 5ISZ | MHCI | GILGFVFTL | AFDTNAGKST | ASSIFGQREQY |
| 6U3N | MHCII | APMPMPELPYP | AVGAGSNYQLI | ASSLEGQGASEQF |
| 6RSY | MHCI | RMFPNAPYL | IGGGTTSGTYKYI | ASSLGFGRDVMR |
| 4MVB | MHCI | QPAEGGFQL | AVSAKGTGSKLS | ASSDAPGQLY |

Table 11: The samples contained in TCRxAI benchmarks (continue table 2)

| PDB | MHC | Peptide | CDRA3 | CDRB3 |
|-----|-----|---------|-------|-------|
| 1MI5 | MHCI | FLRGRAYGL | ILPLAGGTSYGKLT | ASSLGQAYEQY |
| 3VXS | MHCI | RYPLTLGWCF | AVRMDSSYKLI | ASSSWDTGELF |
| 7OW5 | MHCI | VVVGAGGVGK | AMSVPSGDGSYQFT | ASKVGPGQHNSPLH |
| 8GVG | MHCI | RFPLTFGW | AVGFTGGGNKLT | ASSDRDRVPETQY |
| 2VLK | MHCI | GILGFVFTL | AGAGSQGNLI | ASSSRSSYEQY |
| 8GOM | MHCI | RLQSLQTYV | ASSGNTPLV | ASTWGRASTDTQY |
| 1D9K | MHCII | GNSHRGAIEWEGIESG | AATGSFNKLT | ASGGQGRAEQF |
| 6CQQ | MHCII | RFYKTLRAEQASQ | AFKAAGNKLT | ASSRLAGGMDEQF |
| 5HHM | MHCI | GILGLVFTL | AGAGSQGNLI | ASSSSRSSYEQY |
| 4N0C | MHCI | MPAGRPWDL | AVSAKGTGSKLS | ASSDAPGQLY |
| 5C0B | MHCI | RQFGPDFPTI | AMRGDSSYKLI | ASSLWEKLAKNIQY |
| 7PB2 | MHCI | VVVGADGVGK | ALSGPSGAGSYQLT | ASSYGPGQHNSPLH |
| 1MWA | MHCI | EQYKFYSV | AVSGFASALT | ASGGGGTLY |
| 4QRP | MHCI | HSKKKCDEL | ALSDPVNDMR | ASSLRGRGDQPQH |
| 6RPA | MHCI | SLLMWITQV | AVRDINSGAGSYQLT | SVGGSGGADTQY |
| 7N5P | MHCI | SSLCNFRAYV | ILSGGSNYKLT | ASSFFGREQY |
| 7N1F | MHCI | YLQPRTFLL | AVNRDDKII | ASSPDIEQY |
| 5C0C | MHCI | RQFGPDWIVA | AMRGDSSYKLI | ASSLWEKLAKNIQY |
| 6D78 | MHCI | AAGIGILTV | AVNFGGGKLI | ASSWSFGTEAF |
| 4JFF | MHCI | ELAGIGILTV | AVNDGGRLT | AWSETGLGMGGGWQ |
| 4N5E | MHCI | VPYMAEFGM | AVSAKGTGSKLS | ASSDAPGQLY |
| 4JRX | MHCI | LPEPLPQGQLTAY | ALSGFYNTDKLII | ASPGETEAF |
| 7NMF | MHCI | QLPRLFPLL | AEPSGNTGKLI | ASSLHHEQY |
| 3QIW | MHCII | ADLIAYLEQATKG | AAEPSSGQKLV | ASSLNNANSDYT |
| 6ZKX | MHCI | RLPAKAPLLGCG | AVTNQAGTALI | ASSYSIRGSRGEQF |
| 1NAM | MHCI | RGYVYQGL | AMRGDYGGSGNKLI | TCSADRVGNTLY |
| 8PJG | MHCII | PKYVKQNTLKLAR | AVSEQDDKII | ATSDESYGYT |
| 8VCX | MHCII | GQVELGGGPGAESCQ | IVSHNAGNMLT | ASSLERETQY |
| 5YXU | MHCI | KLVALGINAV | AYGEDDKII | ASRRGSAELY |
| 3O4L | MHCI | GLCTLVAML | AEDNNARLM | SARDGTGNGYT |
| 7SG2 | MHCII | QPFPQPEQPFPGS | LVGGLARDMR | SVALGSDTGELF |
| 8GON | MHCI | RLQSLQIYV | ASSGNTPLV | ASTWGRASTDTQY |
| 2JCC | MHCI | ALWGFFPVL | ALFLASSSFSKLV | ASSDWVSYEQY |
| 6G9Q | MHCI | KAPYDYAPI | AALYGNEKIT | ASSDAGGRNTLY |
| 6DKP | MHCI | ELAGIGILTV | AVNFGGGKLI | ASSWSFGTEAF |
| 5NQK | MHCI | ELAGIGILTV | AGGGGADGLT | ASSQGLAGAGELF |
| 2PYE | MHCI | SLLMWITQC | AVRPLLDGTYIPT | ASSYLGNTGELF |
| 6R2L | MHCI | SLSKILDTV | AVGGNDWNTDKLI | ASSPLDVSISSYNEQF |
| 2OI9 | MHCI | QLSPFPFDL | AVSGFASALT | ASGGGGTLY |
| 8F5A | MHCI | TSTLQEQIGW | AVTLNNNAGNMLT | ASSVGGTEAF |
| 7Z50 | MHCII | LQTLALEVEDDPC | AASVRNYKYV | ASSRQGQNTLY |
| 6BGA | MHCII | YVVVPD | AALRATGGNNKLT | ASSLNWSQDTQY |
| 3MV7 | MHCI | HPVGEADYFEY | AVVQDLGTSGSRLT | ASSARSGELF |
| 8VD2 | MHCII | GQVELGGGTPIESC | IVRVAIEGSQGNLI | ASSLRRGDTIY |
| 8VCY | MHCII | GQVELGGGSSPETCI | IVSHNAGNMLT | ASSLERETQY |
| 5YXN | MHCI | KLVALGINAV | AYGEDDKII | ASRRGPYEQY |
| 5E9D | MHCI | ELAGIGILTV | AVTKYSWGKLQ | ASRPGWMAGGVELY |
| 6AMU | MHCI | MMWDRGLGMM | AVNFGGGKLI | ASSLSFGTEAF |
| 5BRZ | MHCI | EVDPIGHLY | AVRPGGAGPFFVV | ASSFNMATGQY |
| 3TJH | MHCI | SPLDSLWWI | AVSAKGTGSKLS | ASSDAPGQLY |
| 3H9S | MHCI | MLWGYLQYV | AVTTDSWGKLQ | ASRPGLAGGRPEQY |
| 4PRP | MHCI | HPVGQADYFEY | AVQDLGTSGSRLT | ASSARSGELF |
| 5IVX | MHCI | RGPGRAFVTI | AASASFGDNSKLI | ASSLGHTEVF |
| 4Y1A | MHCII | LQPLALEGSLQKRG | AASSSAGGTSYGKLT | ASRPRDPVTQY |
| 2IAM | MHCII | GELIGILNAAKVPAD | AALIQGAQKLV | ASTYHGTGY |
| 6U3O | MHCII | AVVQSELPYPEGS | IAFQGAQKLV | ASSFRALAADTQY |

Table 12: The samples contained in TCRxAI benchmarks (continue table 3)

| PDB | MHC | Peptide | CDRA3 | CDRB3 |
|-----|-----|---------|-------|-------|
| 6V0Y | MHCII | GGYAPAKAAAT | ALSDSGSFNKLT | ASSLDWGGQNTLY |
| 5NME | MHCI | SLYNTVATL | AVRTNSGYALN | ASSDTVSYEQY |
| 7T2C | MHCII | TGLAWEWWRTVY | LVGDTGFQKLV | SARDPGGGGSSYEQY |
| 2PXY | MHCII | RGGASQYRPSQ | ALSENYGNEKIT | ASGDASGAETLY |
| 4G9F | MHCI | KRWIIMGLNK | AMRDLRDNFNKFY | ASREGLGGTEAF |
| 3QIB | MHCII | ADLIAYLKQATKG | AALRATGGNNKLT | ASSLNWSQDTQY |
| 4G8G | MHCI | KRWIILGLNK | AMRDLRDNFNKFY | ASREGLGGTEAF |
| 7N2O | MHCI | LRVMMLAPF | AVLSPVQETSGSRLT | ASSVGLFSTDTQY |
| 5TEZ | MHCI | GILGFVFTL | AASFIIQGAQKLV | ASSLLGGWSEAF |
| 3D39 | MHCI | LLFGPVYV | AVTTDSWGKLQ | ASRPGLAGGRPEQY |
| 6ZKW | MHCI | RLPAKAPLL | AVTNQAGTALI | ASSYSIRGSRGEQF |
| 4FTV | MHCI | LLFGYPVYV | AVTTDSWGKLQ | ASRPGLMSAQPEQY |
| 6PX6 | MHCII | APFSEQEQPVLG | AVHTGARLM | ASSHGASTDTQY |
| 6V1A | MHCII | GGYRAPAKAAAT | ALSDSSSFSKLV | ASSLDWASQNTLY |
| 1U3H | MHCII | SRGGASQYRPSQ | AASANSGTYQR | ASGDAGGGYEQY |
| 7N4K | MHCI | SSLENFRRAYV | ILSGGSNYKLT | ASSFFGREQY |
| 2Z31 | MHCII | RGGASQYRPSQ | ALSENYGNEKIT | ASGDASGGNTLY |
| 2ESV | MHCI | VMAPRTLIL | IVVRSSNTGKLI | ASSQDRDTQY |
| 5EUO | MHCI | GILGFVFTL | AGAIGPSNTGKLI | ASSIRSSYEQY |
| 4JFD | MHCI | ELAAIGILTV | AVNDGGRLT | AWSETGLGMGGWQ |
| 6ZKY | MHCI | RLPAKAPL | AVTNQAGTALI | ASSYSIRGSRGEQF |
| 6TRO | MHCI | GVYDGREHTV | VVNHSGGSYIPT | ASSFLMTSGDPYEQY |
| 7N2P | MHCI | GQVMVVAPR | AVSNFNKFY | ASSVATYSTDTQY |
| 7R80 | MHCI | QASQEVKNW | AQLNQAGTALI | ASSYGTGINYGYT |
| 1BD2 | MHCI | LLFGYPVYV | AAMEGAQKLV | ASSYPGGGFYEQY |
| 4L3E | MHCI | ELAGIGILTV | AVNFGGGKLI | ASSWSFGTEAF |
| 7PHR | MHCI | YLEPGPVTV | ATDGSTPMQ | ASSWGAPYEQY |
| 3FFC | MHCI | FLRGRAYGL | AMREDTGNQFY | ASSFTWTSGGATDTQY |
| 4JRY | MHCI | LPEPLPQGQLTAY | AVGGGSNYQLI | ASSRTGSTYEQY |
| 5SWS | MHCI | ASNENMETM | AASEGSGSWQLI | ASSAGLDAEQY |
| 6UZ1 | MHCI | LLFGYPVYV | AVTTDRSGKLQ | ASRPGAAGGRPELY |
| 1FO0 | MHCI | INFDFNTI | AMRGDYGGSGNKLI | TCSADRVGNTLY |
| 7JWI | MHCI | ASNENMETM | AASETSGSWQLI | ASSRDLGRDTQY |
| 8D5Q | MHCI | HPGSVNEFDF | ALGDPTGANTGKLT | TCSAGRGGYAEQF |
| 6VRM | MHCI | HMTEVVRHC | VVQPGGYQKVT | ASSEGLWQVGDEQY |
| 7N2R | MHCI | TRLALIAPK | AVSNFNKFY | ASSVATYSTDTQY |
| 1FYT | MHCII | PKYVKQNTLKLAT | AVSESPFGNEKLT | ASSSTGLPYGYT |
| 3QFJ | MHCI | LLFGFPVYV | AVTTDSWGKLQ | ASRPGLAGGRPEQY |
| 3GSN | MHCI | NLVPMVATV | ARNTGNQFY | ASSPVTGGIYGYT |
| 6V13 | MHCII | GGYRAPAKAAAT | ALSPSNTNKVV | ASSLDWGVNTLY |
| 7OW6 | MHCI | VVVGADGVGK | AMSVPSGDGSYQFT | ASKVGPGQHNSPLH |
| 4OZF | MHCII | APQPELPYPQPGS | IAFQGAQKLV | ASSFRALAADTQY |
| 4JFE | MHCI | ELAGIGALTV | AVNDGGRLT | AWSETGLGMGGWQ |
| 3MV9 | MHCI | HPVGEADYFEY | AVQDLGTSGSRLT | ASSARSGELF |
| 6Q3S | MHCI | SLLMWITQV | AVRPTSGGSYIPT | ASSYVGNTGELF |
| 5MEN | MHCI | ILAKFLHWL | AVDSATSGTYKYI | ASSYQGTEAF |
| 1AO7 | MHCI | LLFGYPVYV | AVTTDSWGKLQ | ASRPGLAGGRPEQY |
| 4H1L | MHCII | QHIRCNIPKRISA | AVGASGNTGKLI | ASSLRDGYTGELF |

Table 13: The samples contained in TCRxAI benchmarks (continue table 4)

## B    REPRODUCIBILITY STATEMENT

The source code is publicly available at our project website (`https://qcai.jiarui.li/`) and on GitHub (`https://github.com/Jiarui0923/QCAI`).

## C    LARGE LANGUAGE MODEL USAGE STATEMENT

We employed large language models (LLMs), primarily ChatGPT, in two limited ways:

- as a coding assistant, and
- for polishing written text.

**Coding Assistant**    LLMs were consulted to clarify documentation, organize API references, and suggest debugging strategies. All code, documentation, and fixes obtained were manually reviewed and verified by the authors.

**Polishing Article**    LLMs were used only to refine the clarity and style of sentences written by the authors and to format tables from raw data. No raw text or substantive content was generated by LLMs. All refined content was manually checked and further revised by the authors.

