# OpenReview forum: "Quantifying Cross-Attention Interaction in Transformers for Interpreting TCR-pMHC Binding"
_ICLR.cc/2026/Conference — ICLR 2026 Poster_

### Official Review · Reviewer_dbL7 · 2025-10-29

**Soundness:** 3
**Presentation:** 3
**Contribution:** 3
**Rating:** 6
**Confidence:** 4

**Summary:**

This paper proposed QCAI to interpret encoder-decoder transformer models for the TCR-pMHC binding task, where existing explainability (XAI) methods are inadequate for the asymmetric nature of cross-attention. It combines gradient-based scoring with a Moore-Penrose pseudoinverse decomposition and element-wise score aggregation to attribute quantitative contributions back to the distinct query and key input tokens. The paper establishes the TCR-XAI benchmark, a curated dataset of 274 experimentally determined protein structures that utilizes atomic distance as a ground truth for interaction importance. Also  included is direct-comparison metric, the Binding Region Hit Rate (BRHR), to confirm QCAI's ability to generate quantitativeuperior and biologically plausible insights.

**Strengths:**

1. The proposed method, which interprets the complex cross-attention mechanism, effectively addresses the application gap between the existing post-hoc explainable AI methods and scientific research problems conceptually.

3. The development of the TCR-XAI benchmark is useful for evaluation, grounding XAI performance in objective, physical atomic distances rather than subjective metrics.

4. A comprehensive validation framework is used that employs a suite of metrics, including ground-truth correspondence, perturbation influence, and the novel Binding Region Hit Rate.

**Weaknesses:**

1. **Ambiguity in Cross-Attention Inputs:** The paper does not clearly define the roles of the input sequences (CDR and Peptide) within the TULIP model's cross-attention mechanism (Figure 1). It is ambiguous which sequence provides the query vectors and which provides the key and value vectors. Given that the TULIP model is autoregressive and can predict any one sequence conditioned on the others, these roles may be dynamic. A detailed visualization of this process, **explicitly tagging each component with its corresponding input entity (e.g., CDR3α as query, Peptide as key)** , would be highly beneficial for interpreting the model's mechanics and the application of the QCAI method.

2. **Clarification of the Core Methodological Distinction:** The description of the gradient-based importance score does not clearly articulate how its computation differs for cross-attention versus self-attention. In particular, the manuscript does not say what is **unique or nontrivial about handling cross-attention compared to standard self-attention**. A more explicit treatment of these differences would help clarify the specific methodological contribution of QCAI, beyond applying generic gradient-weighted attention analysis.

3. **Experimental Setting around Attention Modules:** In the current experiments, the baseline methods are evaluated only on self-attention layers, with cross-attention omitted, whereas QCAI is evaluated on both self-attention and cross-attention. This raises two concerns. First, part of QCAI’s performance gain may simply come from exploiting additional cross-attention information that the baselines are not allowed to use, which makes the comparison potentially unfair. Second, because QCAI is only reported in its combined form, it is not possible to disentangle how much of the improvement is due to self-attention versus cross-attention. It would substantially strengthen the empirical analysis to report ROC-AUC, LOdds, and AOPC for **(i) QCAI applied only to self-attention and (ii) QCAI applied only to cross-attention**, in order to quantify the individual contribution of each attention component.

**Questions:**

Although transformer models offer promising results, its positional encoding scheme is not suited for modelling the positional relations in interacting sequences, because (1) relative distance between residues across sequences are unknown and have to be inferred first; (2) existing positional encoding vectors are not physically meaningful in the first place. Therefore, while it is useful to provide explanations on a complex, data-driven black-box model, it could be more significant to devise a model that itself accord with first principles (or at least provide physically meaningful intermediate results). The author may refer to the recent work entitled "Sliding-attention transformer neural architecture for predicting T cell receptor-antigen-human leucocyte antigen binding" in Nature Machine Intelligence, 2024, 6(10): 1216-1230.

Typos:
Lines 197–198: The phrase ‘denotes calculate max among feature dimension’ is ungrammatical.
Lines 407–408: The phrase ‘critical contacts with peptide peptides’ contains a redundant repetition of ‘peptide.’
Line 805: The text ‘was proposedAbnar & Zuidema’ is missing both a space and the preposition ‘by’.

---

> ### Author Response · Authors · 2025-11-25
>
> We sincerely thank the reviewer for the comments. We have addressed all concerns with additional experiments, clarification, or discussion. References such as `L#` and `Fig.#` indicate changes in the revised manuscript, with updates highlighted in blue.
>
> ## Responses to Weaknesses
> 1. **Role of Input Sequences in TULIP**
> As requested by the reviewer, we have clarified the roles of the epitope peptide, CDR3a, and CDR3b in TULIP, as detailed in `Appendix A.4` (`L946`). Specifically, peptide features serve as the query, while CDR3a and CDR3b features function as the keys and values in the cross-attention mechanism.
>
> 2. **Nontrivial Differences Between Handling Self- and Cross-Attention**
> As requested by the reviewer, we have emphasized this distinction in `Section 2.1` (`L128`), highlighting two main points: (1) the attention matrix of cross-attention fuses information from two modalities, and (2) it is no longer a square matrix, making direct measurement infeasible.
>
> 3. **QCAI on Cross- vs. Self-Attention**
> As requested by the reviewer, we conducted an ablation study by applying QCAI only to cross- and self-attention, as reported in `Appendix A.11`. The results indicate that QCAI applied to cross-attention is the primary contributor to the final explanation and cross-attention plays a significant role in transformers that incorporate cross-attention. We will add this discussion to Results section.
>
> ## Responses to Questions
> **Transformers and Positional Encodingfor TCR-pMHC Binding**
> The author rightly points out the limitations of sequence-based transformer architectures. However when we consider experimental data available for training for TCR-pMHC binding, there are 3 orders of magnitude more sequencing-based training examples (~200K) than TCR-pMHC complex structures (<300). The goal of QCAI is to shed light on why sequence-based architectures achieve excellent performance, and to provide insights though explainability as to how they can be improved (e.g. with structural information). Specifically the approach taken by PISTE is interesting and while QCAI is not directly applicable, our methods (e.g. attention matrix decomposition to extract cross-attention contributions) can be used to potentially improve performance. An important future direction that we are currently exploring is to develop models that are explain-by-design models that effectively use both experimental sequence and experimental structure data. We have added this discussion to `Section 5.1` (`L536`) as part of our future work.
>
>
> ### Typo Fixing
> 1. **Ungrammatical Sentence**
> `L97` We have re-written this sentence for clarity.
>
> 2. **Redundant Word "Peptide"**
> `L408` We have corrected "peptide peptide" to "epitope peptide".
>
> 3. **Missing Space and "by"**
> `L859` We have added the missing space and changed the phrase to "proposed by".

---

> > ### Comment · Reviewer_dbL7 · 2025-11-26
> > **thanks for your responses**
> >
> > Thanks for your clarifications. I have read through the responses and I tend to believe that my previous rating was appropriate.

---

### Official Review · Reviewer_XjQm · 2025-10-30

**Soundness:** 4
**Presentation:** 4
**Contribution:** 3
**Rating:** 6
**Confidence:** 5

**Summary:**

This work presents a method for the interpretation of TCR-pMHC binding prediction models with encoder-decoder structure and cross attention mechanisms, as well as a benchmark dataset of ground truth interacting residues defined by residue distance.. Specifically, the proposed method, quantifying cross-attention interaction (QCAI), combines a GradCAM-style intrinsic importance and a relevance score derived from the attention matrix. The authors demonstrated superior performance of the proposed importance in the task of binding site retrieval by log-odds and area under perturbation metrics.

**Strengths:**

1\. This work offers insights into the important problem of interaction interpretation in immune-proteins. Considering the scarcity of data and limited prediction performance of existing models, a reliable interpretation method would allow maximal use of available data and models. The method and findings could potentially guide rational design of TCRs and respective immunotherapy.

2\. The authors provide solid theoretical justifications of the methodology as well as practical insights.

3\. The work offers a useful benchmark dataset for future work.

4\. Case studies with visualization clearly demonstrate the biochemical implications of the proposed importance scores.

**Weaknesses:**

Despite the sound problem setup and results, my major concern is the limited application both within and beyond the domain of TCR-pMHC. Specifically:

1\. The scope of the defined task and the respective benchmark dataset is somewhat limited. Distance is not the only indicator of interaction and only weakly indicates "importance" overall, considering residue contributions to TCR-pMHC interactions are somewhat additive (smaller, weaker interactions than dominating hotspots). Though that may be hard to analyze with the current tools, it should be briefly discussed.

2\. The method relies on TCR-pMHC structures, which is rather rare compared to other interaction types.

3\. TCR-pMHC is an important and complex subject, the proposed method could be widely applied to many other interaction types. Generalizability needs to be shown, or at least discussed.

**Questions:**

1\. Since the experiments are only performed on known, "positive" TCR-pMHC pairs (those already with a stable structure), what bias might that introduce? Please discuss.

2\. Could the proposed method be applied to downstream prediction tasks such as impacts of sequence or structure perturbations, or other protein-protein interaction types?  Please discuss.

3\. Since the method requires 3D structure of the interaction complex, how would predicted structures (such as by AlphaFold base model or fine-tuned on TCR-pMHC) be used when structures are not available? From another perspective, is it possible to use the method for the evaluation of predicted structures or unknown interaction pairs?

4\. Does the range and scale of the residue importance show any relationship to the TULIP model's confidence of prediction?

5\. Since some samples in PDB are close neighbours (differing by only one or two residues), such as 2PXY and 2Z3, does the importance score show any different or similar patterns between the different residues?

---

> ### Author Response · Authors · 2025-11-25
>
> We sincerely thank the reviewer for the comments. We have addressed all concerns with additional experiments, clarification, or discussion. References such as `L#` and `Fig.#` indicate changes in the revised manuscript, with updates highlighted in blue.
>
> ## Responses to Weaknesses
> 1. **Discussion of Binding Contributions**
> The reviewer is correct to point out that binding contributions are additive. QCAI provides importance values for all residues in the input sequences; we show in our case studies how this differs based on the specific inputs. Our BRHR metric uses a threshold on distance to provide a concrete (i.e., conservative) metric on explanation quality; `Appendix A.8` shows results for a variety thresholds. Any comprehensive evaluation requires a thresholded approach, but another interesting avenue for evaluation (as pointed out by another reviewer) is to use energetics rather than distance. We are exploring this currently and have added a discussion in `Section 5.1` (`L524`) on this point.
>
> 2. **Applicability of QCAI to Other Applications**
> As requested by the reviewer, we discuss how QCAI can be used for other applications in `Section 5.1` (`L528`). First, we discuss other interaction types here. Given the emergence of several cross-attention models for protein-protein interactions and immunological tasks, such as PALM-H3 [1] for antigen generation, UniPMT [2] for peptide-MHC prediction, ProtAttBA [3] for antibody-antigen prediction, and HBFormer[4] for human-virus interaction identification, QCAI provides a method for opening the black box of cross-attentions in these models and revealing their underlying mechanisms.
> We also provide a simple case study in which we apply QCAI to analyze CLIP with cross-attention, a vision-language model. As shown in `Appendix Figure 14` and `Appendix A.12`, QCAI can identify interactions between the text and image modalities in cross-attention and highlight the relative importance of the image and text for a given classification label.
>
> [1] He, et. al. (2024). De novo generation of SARS-CoV-2 antibody CDRH3 with a pre-trained generative large language model. Nature Communications, 15(1), 6867.
> [2] Zhao, et. al. (2025). A unified deep framework for peptide–major histocompatibility complex–T cell receptor binding prediction. Nature Machine Intelligence, 1-11.
> [3] Liu, et. al. (2025). Sequence-only prediction of binding affinity changes: a robust and interpretable model for antibody engineering. Bioinformatics, 41(8), btaf446.
> [4] Zhang, et. al. (2024). HBFormer: a single-stream framework based on hybrid attention mechanism for identification of human-virus protein–protein interactions. Bioinformatics, 40(12), btae724.
>
>
> ## Responses to Questions
> 1. **Class Imbalance in the TCR-XAI Benchmark**
> The reviewer raised concerns about potential bias from using only positive TCR-pMHC pairs. We note that the TCR-XAI benchmark is meant to be used to evaluate explainability for positive examples. Modeling of non-binding (i.e. unfavorable interactions between TCR and pMHC) could be a very interesting avenue of work. QCAI can be used to explain negative predictions, but to evaluate explainability we would need ground-truth negative interactions. These would require not just negative examples but also a comprehensive physicochemical evaluation of "anti-binder" residues; we have not found studies in the literature.
>
> 2. **Evaluation of Unknown Interactions or Predicted Structures**
> Regarding the reviewer's question about applying QCAI to evaluate unknown interactions or predicted structures, we agree this is a promising direction. While most TCR-pMHC binding data are sequence-based, QCAI can still provide structural insights. As demonstrated in our structural case study, QCAI may help researchers design experiments and investigate underlying mechanisms. Additionally, QCAI can be used to assess the consistency between its inferred importance patterns and predicted TCR-pMHC structures, potentially revealing structural inaccuracies or areas requiring further validation.
>
> 3. **Relationship Between TULIP Prediction Confidence and Importance**
> As requested by the reviewer, we investigated the correlation between prediction confidence and importance, as discussed in `Appendix A.8` (`L1206`). Using both MixTCRpred and TULIP, we found that samples with lower prediction confidence exhibit reduced explanation quality compared to high-confidence samples.
>
> 4. **The Difference of Residue Importance Between Similar PDBs**
> The reviewer asked how importance patterns change across similar TCR-pMHC structures. To address this, we added a case study on 2PXY and 2Z31, two closely related structures, in `Section 4.4` (`L503`). The results show that QCAI consistently identifies key contact regions despite minor sequence differences, though such differences can still influence overall explanation quality.

---

> > ### Comment · Reviewer_XjQm · 2025-11-28
> >
> > I appreciate the authors' detailed responses. I'd like to maintain my current scores.

---

### Official Review · Reviewer_rFG4 · 2025-11-01

**Soundness:** 3
**Presentation:** 3
**Contribution:** 2
**Rating:** 6
**Confidence:** 3

**Summary:**

Transformer-based models, while effective for predicting TCR-pMHC binding, are "black boxes" because existing explainable AI methods are not designed for their encoder-decoder architectures. This paper introduces Quantifying Cross-Attention Interaction, a post-hoc explanation method specifically designed to interpret the cross-attention mechanisms in the decoder part of the transformer. To quantitatively evaluate this new method, the authors also compiled TCR-XAI, a benchmark of 274 TCR-PMHC crystal structures derived from STCRDab and TCR3d 2.0 that uses physical residue distance as a ground truth for binding interaction. The authors demonstrate that QCAI achieves good performance on this benchmark across multiple metrics, including ROC-AUC, perturbation studies (LOdds and AOPC), and a new metric called Binding Region Hit Rate.

**Strengths:**

- The primary strength is providing an explainable AI method for encoder-decoder transformers, like TULIP, which current methods designed for encoder-only models cannot adequately interpret.
- QCAI's performance is validated against a suite of competing methods (e.g., AttnLRP, TokenTM, Rollout) using ROC analysis, two different perturbation metrics (AOPC and LOdds) , and the authors' own BRHR metric, demonstrating SOTA results across most of them.

**Weaknesses:**

- The paper explicitly states that QCAI is up to 50x slower per sample than other methods due to the necessary pseudo-inverse operations.
- The new TCR-XAI benchmark is heavily skewed towards MHC-I samples, which may limit the generalizability of the findings for MHC-II complexes.
- The benchmark relies on atomic distance as a "proxy for ground-truth importance". An assessment based on an energy function would have been more appropriate

**Questions:**

- In the introduction, TULIP is mentioned as an example of a BERT-style model, but as mentioned in other parts of the paper, TULIP is an autoregressive encoder-decoder model, not an encoder-only BERT-like model.
- The 2017 transformer paper should be cited in section 2
- Did the author consider evaluating their method on a transformer trained on natural language?

---

> ### Author Response · Authors · 2025-11-25
>
> We sincerely thank the reviewer for the comments. We have addressed all concerns with additional experiments, clarification, or discussion. References such as `L#` and `Fig.#` indicate changes in the revised manuscript, with updates highlighted in blue.
>
> ## Responses to Weaknesses
> 1. **Cost of Pseudo-Inverse Computation**
> As noted by the reviewer, QCAI is slower approximately 50x due to the use of the pseudo-inverse. However, our current implementation is naive due to the tolerable overall cost. We note that techniques such as low-rank approximation [1] can substantially accelerate pseudoinverse computation and reduce computational cost from $O(n m^2)$ to be $O(log(nm))$, where $m$ is the larger dimension of the matrix and $n$ the smaller. These algorithmic optimizations can be incorporated in future applications to reduce computational overhead.
>
> [1] Lee, N., & Cichocki, A. (2016). Regularized computation of approximate pseudoinverse of large matrices using low-rank tensor train decompositions. SIAM Journal on Matrix Analysis and Applications, 37(2), 598-623.
>
> 2. **MHC-I Skewed TCR-XAI Benchmark**
> We agree that TCR-XAI is skewed toward MHC-I, but this reflects the limited availability of MHC-II structural data. Balancing the benchmark would drastically reduce its size. While this imbalance may affect generalizability to MHC-II complexes, its impact on explainability evaluation is limited, as existing models are also predominantly trained on MHC-I skewed dataset.
>
> 3. **Utilizing Energy Function as a Proxy for Ground-Truth Importance**
> The reviewer poses an interesting idea as residue-level energetics incorporate physicochemical information that may provide additional detail about binding contributions. We attempted a simple implementation using PyRosetta but could not obtain the necessary inter-unit energy terms for an appropriate implementation. We added a discuss in `Section 5.1` (`L514`) to use energy function as an indicator.
>
>
> ## Responses to Questions
> 1. **TULIP Is Not an Encoder Only Model**
> `L47` We have removed the confusing description in the Introduction section.
>
> 2. **Transformer Paper Citation**
> `L97` We have added a citation to the original paper introducing transformers (Vaswani et al., 2017) in `Section 2`.
>
> 3. **Application of QCAI to Other Domains**
> As requested by the reviewers, we conducted a simple case study applying QCAI to analyze CLIP with cross-attention, a vision-language model. As shown in `Appendix Figure 15` and `Appendix A.13`, QCAI can identify interactions between the text and image modalities in cross-attention and highlight the relative importance of the image and text for a given classification label.

---

> > ### Comment · Reviewer_rFG4 · 2025-11-27
> > **thank you for your response**
> >
> > I appreciate the response and updates to the paper, especially the addition of section A.13. I will maintain my initial score.

---

### Official Review · Reviewer_iex5 · 2025-11-01

**Soundness:** 3
**Presentation:** 3
**Contribution:** 2
**Rating:** 4
**Confidence:** 2

**Summary:**

The paper introduces a novel method to interpret the features learned by an encoder-decoder transformer with cross attention. The method relies on the evaluation of attention scores for query and key values, and aggregates them across the layers of the decoder. The method is then tested against others on the task of predicting contact residues from models trained on protein binding (TCR-pMHC), showing increase in performance. In order to do so, the authors provide an annotated benchmark dataset.

**Strengths:**

Methods to explain cross-attention in transformers are needed to go beyond their use as “black boxes”. The tested case is timely and interesting for the community working in molecular biology. The benchmark dataset curated by the authors can also be useful.

**Weaknesses:**

If I understand correctly, the task in which the authors test their method is rather hard: all the competitors perform in a way that is comparable or worse than random chance (Fig. 2), showing biased assessment. For a method which is “potential to be applied to other fields”, as the authors claim, other applications, where explainable methods work more reliably, should be tested.

**Questions:**

- I understand the the method proposed outperforms the others, but how should I read the overall low AUCs in Fig. 2? Are most methods performing worse than random chance? Can the authors comment on why they consider an AUC of .6 as “demonstrating strong alignment” between importance scores and binding interactions?
- Can the authors test their method on tasks where explainable approaches work more convincingly?
- The method is applied to TULIP, a state-of-the-art model for TCR-pHMC binding prediction. Are the benchmark data curated by the authors in the training set of TULIP, or not?

Minor:

The notation for element-wise multiplication is not consistent throughout the paper ($\odot$ vs. $\cdot$)

---

> ### Author Response · Authors · 2025-11-25
>
> We sincerely thank the reviewer for the comments. We have addressed all concerns with additional experiments, clarification, or discussion. References such as `L#` and `Fig.#` indicate changes in the revised manuscript, with updates highlighted in blue.
>
> 1. **Random Baseline Performance**
> In response to the reviewer’s question regarding why some competitors perform worse than “random,” we clarify the following. First, this is a challenging task, as the models have no access to structural information. Second, unlike binary TCR-pMHC binding prediction, this task requires identifying contacting residues, where random guessing can naturally yield worse-than-random performance. In addition, competing methods cannot access cross-attention and therefore miss critical information. As an evidence, GradCAM, applied only to the last layer, benefits from direct gradient information and consequently performs better than other baselines.
>
> 2. **Application of QCAI to Other Domains**
> While a full demonstration of QCAI in another domain would require a substantial time commitment, we did implement a proof-of-concept applicaiton in which QCAI is applied to analyze the grounding performance of CLIP with cross-attention, a vision-language model. As shown in `Appendix Figure 14` and `Appendix A.12`, QCAI can identify interactions between the text and image modalities in cross-attention and highlight the relative importance of the image and text for a given classification label. This is similar to identify the binding elements across two inputs. We also added `Section 5.1` to discuss various applications as future work.
>
> 3. **TCR-XAI Benchmark**
> As requested by the reviewer, we clarify that the TCR-XAI benchmark is not curated from the TULIP training dataset. It includes 176 distinct epitopes, none appearing in more than 3.3% of samples (`Appendix A.9`, `L1248`). We also compared BRHR across samples grouped by Levenshtein distance to the TULIP training set (`Appendix Table 5`), showing that QCAI's performance remains stable even as samples diverge.
>
> ### Minor
> **Inconsistent Element-wise Product Notation**
> `L177` The symbols $\odot$ and $\cdot$ denote element-wise product and matrix product, respectively. We have clarified this in the manuscript.

---

### Author Response · Authors · 2025-11-21

We sincerely thank the reviewers for their detailed comments. We have addressed all concerns with additional experiments, clarification, or discussion. Below is an initial overview of our responses to the major points that we identified in reviewer comments. **Detailed, point-by-point replies were added for each reviewer questions.** References such as `L#` and `Fig.#` indicate changes in the revised manuscript, with updates highlighted in blue.

## Major Comments
These are the major concerns we identified from reviewer feedback. We have addressed them by clarifying the questions and/or adding experiments to support our responses.

1. **TCR-XAI Benchmark**
As requested by the reviewer, we clarify that the TCR-XAI benchmark is not curated from the TULIP training dataset. It includes 176 distinct epitopes, none appearing in more than 3.3% of samples (`Appendix A.9`, `L1248`). We also compared BRHR across samples grouped by Levenshtein distance to the TULIP training set (`Appendix Table 5`), showing that QCAI's performance remains stable even as samples diverge.

2. **Application of QCAI to Other Domains**
While a full demonstration of QCAI in another domain would require a substantial time commitment, we did implement a proof-of-concept applicaiton in which QCAI is applied to analyze the grounding performance of CLIP with cross-attention, a vision-language model. As shown in `Appendix Figure 14` and `Appendix A.12`, QCAI can identify interactions between the text and image modalities in cross-attention and highlight the relative importance of the image and text for a given classification label. This is similar to identify the binding elements across two inputs. We also added `Section 5.1` to discuss various applications as future work.

3. **Utilizing Energy Function as a Proxy for Ground-Truth Importance**
The reviewer poses an interesting idea as residue-level energetics incorporate physicochemical information that may provide additional detail about binding contributions. We attempted a simple implementation using PyRosetta but could not obtain the necessary inter-unit energy terms for an appropriate implementation. We added a discuss in `Section 5.1` (`L522`) to use energy function as an indicator.

4. **Relationship Between TULIP Prediction Confidence and Importance**
As requested by the reviewer, we investigated the correlation between prediction confidence and importance, as discussed in `Appendix A.8` (`L1206`). Using both MixTCRpred and TULIP, we found that samples with lower prediction confidence exhibit reduced explanation quality compared to high-confidence samples.

5. **QCAI on Cross- vs. Self-Attention**
As requested by the reviewer, we conducted an ablation study by applying QCAI only to cross- and self-attention, as reported in `Appendix A.11`. The results indicate that QCAI applied to cross-attention is the primary contributor to the final explanation and cross-attention plays a significant role in transformers that incorporate cross-attention. We will add this discussion to Results section.

6. **Nontrivial Differences Between Handling Self- and Cross-Attention**
As requested by the reviewer, we have emphasized this distinction in `Section 2.1` (`L128`), highlighting two main points: (1) the attention matrix of cross-attention fuses information from two modalities, and (2) it is no longer a square matrix, making direct measurement infeasible.

7. **Role of Input Sequences in TULIP**
As requested by the reviewer, we have clarified the roles of the epitope peptide, CDR3a, and CDR3b in TULIP, as detailed in `Appendix A.4` (`L946`). Specifically, peptide features serve as the query, while CDR3a and CDR3b features function as the keys and values in the cross-attention mechanism.

8. **Random Baseline Performance**
In response to the reviewer's question regarding why some competitors perform worse than "random," we clarify the following. First, this is a challenging task, as the models have no access to structural information. Second, unlike binary TCR-pMHC binding prediction, this task requires identifying contacting residues, where random guessing can naturally yield worse-than-random performance. In addition, competing methods cannot access cross-attention and therefore miss critical information. As an evidence, GradCAM, applied only to the last layer, benefits from direct gradient information and consequently performs better than other baselines.

---

> ### Author Response · Authors · 2025-11-21
>
> _[ Continue ]_
>
> 9. **Evaluation of Unknown Interactions or Predicted Structures**
> Regarding the reviewer's question about applying QCAI to evaluate unknown interactions or predicted structures, we agree this is a promising direction. While most TCR-pMHC binding data are sequence-based, QCAI can still provide structural insights. As demonstrated in our structural case study, QCAI may help researchers design experiments and investigate underlying mechanisms. Additionally, QCAI can be used to assess the consistency between its inferred importance patterns and predicted TCR-pMHC structures, potentially revealing structural inaccuracies or areas requiring further validation.
>
> 10. **MHC-I Skewed TCR-XAI Benchmark**
> We agree that TCR-XAI is skewed toward MHC-I, but this reflects the limited availability of MHC-II structural data. Balancing the benchmark would drastically reduce its size. While this imbalance may affect generalizability to MHC-II complexes, its impact on explainability evaluation is limited, as existing models are also predominantly trained on MHC-I skewed dataset.
>
> 11. **Cost of Pseudo-Inverse Computation**
> As noted by the reviewer, QCAI is slower approximately 50x due to the use of the pseudo-inverse. However, our current implementation is naive due to the tolerable overall cost. We note that techniques such as low-rank approximation [1] can substantially accelerate pseudoinverse computation and reduce computational cost from $O(n m^2)$ to be $O(log(nm))$, where $m$ is the larger dimension of the matrix and $n$ the smaller. These algorithmic optimizations can be incorporated in future applications to reduce computational overhead.
>
> [1] Lee, N., & Cichocki, A. (2016). Regularized computation of approximate pseudoinverse of large matrices using low-rank tensor train decompositions. SIAM Journal on Matrix Analysis and Applications, 37(2), 598-623.
>
> 12. **The Difference of Residue Importance Between Similar PDBs**
> The reviewer asked how importance patterns change across similar TCR-pMHC structures. To address this, we added a case study on 2PXY and 2Z31, two closely related structures, in `Section 4.4` (`L503`). The results show that QCAI consistently identifies key contact regions despite minor sequence differences, though such differences can still influence overall explanation quality.
>
> 13. **Transformers and Positional Encodingfor TCR-pMHC Binding**
> The author rightly points out the limitations of sequence-based transformer architectures. However when we consider experimental data available for training for TCR-pMHC binding, there are 3 orders of magnitude more sequencing-based training examples (~200K) than TCR-pMHC complex structures (<300). The goal of QCAI is to shed light on why sequence-based architectures achieve excellent performance, and to provide insights though explainability as to how they can be improved (e.g. with structural information). Specifically the approach taken by PISTE is interesting and while QCAI is not directly applicable, our methods (e.g. attention matrix decomposition to extract cross-attention contributions) can be used to potentially improve performance. An important future direction that we are currently exploring is to develop models that are explain-by-design models that effectively use both experimental sequence and experimental structure data. We have added this discussion to `Section 5.1` (`L536`) as part of our future work.
>
> ## Minor Comments
> These comments concern clarity, vocabulary, and syntax. We have addressed them either by correcting the manuscript or by providing clarifications.
> 1. **Inconsistent Element-wise Product Notation**
>    `L177` The symbols $\odot$ and $\cdot$ denote element-wise product and matrix product, respectively. We have clarified this in the manuscript.
> 2. **Transformer Paper Citation**
>    `L97` We have added a citation to the original paper introducing transformers (Vaswani et al., 2017) in `Section 2`.
> 3. **TULIP Is Not an Encoder Only Model**
>    `L47` We have removed the confusing description in the Introduction section.
> 4. **Ungrammatical Sentence**
>    `L97` We have re-written this sentence for clarity.
> 5. **Redundant Word "Peptide"**
>    `L408` We have corrected "peptide peptide" to "epitope peptide".
> 6. **Missing Space and "by"**
>    `L859` We have added the missing space and changed the phrase to "proposed by".

---

### Author Response · Authors · 2025-12-01
**Summary for AC**

Dear AC, SAC, and PC,

We acknowledge the current situation and sincerely appreciate your and the reviewers’ hard work. To support the AC in evaluating our submission, **we have summarized our paper’s contributions as well as the main concerns raised by the reviewers, along with our corresponding responses**. We believe that all concerns have been thoroughly addressed. Reviewers `rFG4`, `XjQm`, and `dbL7` have reviewed our responses and maintained their positive ratings. Although reviewer `iex5` has not replied, we have responded to and resolved each of their concerns individually. Additional details and point-by-point responses are provided below and in the revised manuscript.

## Our Paper's Contribution
We propose **Quantifying Cross-Attention Interaction (QCAI)**, a new post-hoc method for interpreting cross-attention mechanisms in transformer decoders for TCR-pMHC binding prediction. We also introduce **TCR-XAI**, a benchmark of 274 experimentally determined TCR-pMHC structures that uses residue-level physical distances as proxy ground-truth explanations. Under this benchmark, QCAI achieves state-of-the-art performance in both interpretability and predictive accuracy. Through case studies, we further show that QCAI can identify biologically meaningful residues. Finally, we apply QCAI to the cross-attention between CLIP’s text and image encoders, demonstrating its potential generality across domains.

## Responses to Main Concerns
1. **Application of QCAI to Other Domains**
We implemented a proof-of-concept application analyzing the grounding performance of CLIP with cross-attention. As shown in `Appendix Figure 14` and `Appendix A.12`, QCAI identifies text-image interactions in cross-attention and highlights the relative contribution of each modality to a given label, analogous to detecting binding elements across two inputs. We also expanded `Section 5.1` to discuss broader future applications.
2. **TCR-XAI Benchmark**
As requested by the reviewer, we clarify that the TCR-XAI benchmark is not curated from the TULIP training dataset. It includes 176 distinct epitopes, none appearing in more than 3.3% of samples (`Appendix A.9`, `L1248`). We also compared BRHR across samples grouped by Levenshtein distance to the TULIP training set (`Appendix Table 5`), showing that QCAI's performance remains stable even as samples diverge.
3. **Discussion of Binding Contributions**
The reviewer is correct that binding contributions are additive. QCAI provides residue-level importance scores, and our case studies show how these vary with different inputs. Our BRHR metric uses a distance threshold as a concrete, conservative evaluation, with results across thresholds in `Appendix A.8`. While thresholding is necessary, using energetic measures is a promising alternative; we are exploring this and have added a discussion in `Section 5.1` (`L524`).
4. **Random Baseline Performance**
Regarding why some baselines perform worse than “random,” this task is inherently difficult because the models lack structural information and must identify specific contacting residues, where random guessing can naturally underperform. In addition, competing methods cannot access cross-attention and thus miss key interaction information. GradCAM, which directly uses gradients from the final layer, retains this information and therefore performs better than other baselines.

We sincerely thank the reviewers, AC, SAC, and PC for your constructive comments, careful consideration, and prompt solution to the current situation. Please let us know if any further questions arise.

Best,
Authors

---

### Meta-Review · Area_Chair_MGMp · 2026-01-17

**Summary:**

This paper proposes Quantifying Cross-Attention Interaction (QCAI), a post-hoc interpretability method aimed at making cross-attention in encoder–decoder Transformers more transparent. The work is motivated by TCR–pMHC binding prediction (e.g., TULIP): QCAI uses gradient signals together with a pseudo-inverse–style decomposition to attribute interaction importance between encoder-side tokens (pMHC) and decoder-side tokens (TCR). A second major contribution is the TCR-XAI benchmark, built from 274 experimentally resolved TCR–pMHC crystal structures, where atomic-distance–based proximity is used as a quantitative proxy ground truth for evaluating explanation quality. Empirically, QCAI performs well across several protocols (ROC-AUC against distance-derived labels, perturbation tests, and the proposed Binding Region Hit Rate / BRHR), and it generally outperforms baselines that are primarily designed around self-attention rather than cross-attention. The case studies are biologically plausible, and the appendix includes a small CLIP-based proof-of-concept that hints at broader applicability beyond the TCR domain.

**Reviewer Concerns:**

Reviewers identified several key issues:
(1) limited demonstration of generalizability beyond the specialized TCR–pMHC setting; (2) potential unfairness in baseline comparisons, as many competing methods are not designed to process cross-attention, potentially inflating QCAI's relative gains; (3) the empirically driven, heuristic character of QCAI (e.g., aggregation via maximum operations and pseudo-inverse approximations, lacking formal faithfulness guarantees); (4) characteristics of the TCR-XAI benchmark, including possible overlap with model training data, MHC-I predominance, and whether atomic distance is an appropriate primary proxy for “interaction importance” (as opposed to contactness); and (5) insufficient clarity on architectural details of TULIP and QCAI's distinct handling of gradients in cross- versus self-attention.

Well-addressed concerns：
Overall, the rebuttal and revisions substantially improve clarity and address a number of concrete concerns, especially around benchmark construction and method description. On (5), the encoder–decoder setup and the roles of query/key/value in cross-attention are described much more clearly, which makes it easier to see what QCAI is attributing and where gradients enter. On (4) (overlap / MHC-I bias), the authors provide additional checks arguing against training-data overlap, include robustness analyses (e.g., sequence-distance style controls), and are upfront that the MHC-I skew largely reflects current structural-data availability rather than selective curation.
For (2), the authors’ framing is mostly reasonable: many popular attribution baselines were developed around self-attention and do not naturally extend to cross-attention, and QCAI’s advantage is in part that it targets exactly this gap. The added ablations supporting cross-attention’s importance help make that point credible. That said, the comparison is still not fully “like-for-like” across identical attribution objects, so the baseline story ultimately depends on accepting that cross-attention interaction attribution is the right target—something that does seem well motivated in this immune receptor–ligand setting.
For (1), the added appendix proof-of-concept on CLIP is helpful as a sanity check that the technique is not intrinsically tied to TCR–pMHC. I would still view this as mitigating rather than fully resolving the generalization concern, because it does not directly address transfer to closely related biological interaction problems (e.g., antigen–antibody binding or broader PPI interface settings) where similar cross-attention structures could be expected to matter. Minor presentation issues (typos, notation) also appear to be cleaned up.

Remaining concerns：
Concerns (3) and parts of (4) remain only partially resolved. On (3), the authors give reasonable intuition for design choices (pseudo-inverse decomposition, conservative max aggregation) and provide some empirical support, but the method still reads as primarily empirically justified rather than grounded in stronger axiomatic properties (e.g., faithfulness / conservation-style guarantees as in some LRP variants). This is common in post-hoc XAI, so I do not see it as disqualifying; it mainly limits how strong the theoretical claims can be and may leave the most theory-oriented reviewer somewhat unconvinced.
On (4) (proxy suitability / absolute-score interpretation), atomic distance is a sensible and widely used proxy for physical contact, and the authors’ justification is plausible. However, contactness is not equivalent to biochemical “importance” (e.g., energetic contribution, solvent-mediated effects, indirect stabilization), so treating distance-threshold labels as a ground truth for importance can only support a weaker claim: alignment to structural contacts. In that light, the moderate absolute AUC values (~0.55–0.60) are not necessarily surprising, but they are also hard to calibrate: it remains unclear what ceiling one should expect under this proxy, and the rebuttal does not add a complementary validation signal. Reviewers’ suggestions such as energy-based proxies are important here, and the current response mostly defers them to discussion/future work without materially strengthening the quantitative grounding. Net: these points feel more like scope/rigor limitations than critical flaws, and they do not undercut the headline contributions (a cross-attention–specific interpretability method plus a valuable structure-based benchmark). I still lean accept.

**Reviewer Scores:**

R1 (initial: 4): Given substantive responses to concerns on generalizability and benchmark validity, likely to increase to 5 or 6 with full discussion engagement.
R2 (initial: 6): Likely unchanged at 6; all raised points were satisfactorily resolved.
R3 (initial: 6): Likely unchanged at 6; detailed methodological queries were addressed, though underlying reservations on theoretical rigor may persist.
R4 (initial: 6): Likely unchanged at 6; clarity enhancements were effective, but heuristic aspects remain a subtle point of caution.

---

### Decision · Program_Chairs · 2026-01-26

Accept (Poster)